# The Panaceas for Improving Low-Rank Decomposition in Communication-Efficient Federated Learning

Shiwei Li [1 2 *]  Xiandi Luo [1 *]  Haozhao Wang [1]  Xing Tang [2]  Shijie Xu [3]  Weihong Luo [3]  Yuhua Li [1]
Xiuqiang He [2]  Ruixuan Li [1]

## Abstract

To improve the training efficiency of federated learning (FL), previous research has employed low-rank decomposition techniques to reduce communication overhead. In this paper, we seek to enhance the performance of these low-rank decomposition methods. Specifically, we focus on three key issues related to decomposition in FL: what to decompose, how to decompose, and how to aggregate. Subsequently, we introduce three novel techniques: Model Update Decomposition (MUD), Block-wise Kronecker Decomposition (BKD), and Aggregation-Aware Decomposition (AAD), each targeting a specific issue. These techniques are complementary and can be applied simultaneously to achieve optimal performance. Additionally, we provide a rigorous theoretical analysis to ensure the convergence of the proposed MUD. Extensive experimental results show that our approach achieves faster convergence and superior accuracy compared to relevant baseline methods. The code is available at https://github.com/Leopold1423/fedmud-icml25.

## 1. Introduction

Federated learning (FL) is a distributed training framework that preserves data privacy by exchanging model parameters instead of sharing decentralized data (McMahan et al., 2017). In an FL system, a central server manages the training process across multiple distributed clients, typically involving the following steps: (1) the server broadcasts the global model to the clients; (2) each client optimizes the model using its local data; (3) after local training, each client sends the model updates (i.e., changes to model parameters) back to the server; and (4) the server aggregates the model updates from all clients to generate a new global model. These steps together constitute one training round in FL.

However, FL generally requires multiple training rounds to achieve convergence, leading to considerable communication overhead. Specifically, communication occurs in two phases: the uplink, where clients transmit local model updates to the server, and the downlink, where the server sends global model parameters to the clients. Latency in both phases can significantly reduce training efficiency, especially in scenarios with limited bandwidth or large model sizes (Hönig et al., 2022). To mitigate this issue, various strategies have been proposed to compress either the uplink (Li et al., 2024a;b; Reisizadeh et al., 2020) or the bidirectional (Tang et al., 2019; Dorfman et al., 2023) communication. This paper focuses on investigating bidirectional compression methods for better training efficiency.

Low-rank decomposition (Sainath et al., 2013) is an effective technique for parameter compression, which approximates a matrix by the product of smaller sub-matrices. It has been widely employed in FL for bidirectional communication compression (Yao et al., 2021; Hyeon-Woo et al., 2022). In these approaches, the server transmits a low-rank model to the clients for training and subsequently receives the optimized models from them. For example, FedHM (Yao et al., 2021) generates a low-rank model by applying truncated singular value decomposition (SVD) to the global model. However, the SVD process introduces approximation errors, causing the low-rank model to deviate from the global model. In contrast, FedLMT (Liu et al., 2024) directly trains a pre-decomposed global model, eliminating the need for SVD. Similarly, FedPara (Hyeon-Woo et al., 2022) trains a pre-decomposed model and enhances the rank of the recovered matrices using the Hadamard product. (Mei et al., 2022) further improves compression by sharing low-rank matrices across multiple layers. These methods have laid the foundation for low-rank decomposition techniques in communication-efficient FL (CEFL). Nevertheless, several challenges remain. In this paper, we address three key

---

*Equal contribution. [1]Huazhong University of Science and Technology, Wuhan, China [2]Shenzhen Technology University, Shenzhen, China [3]FiT, Tencent, Shenzhen, China. Correspondence to: Ruixuan Li <rxli@hust.edu.cn>.

*Proceedings of the 42nd International Conference on Machine Learning*, Vancouver, Canada. PMLR 267, 2025. Copyright 2025 by the author(s).

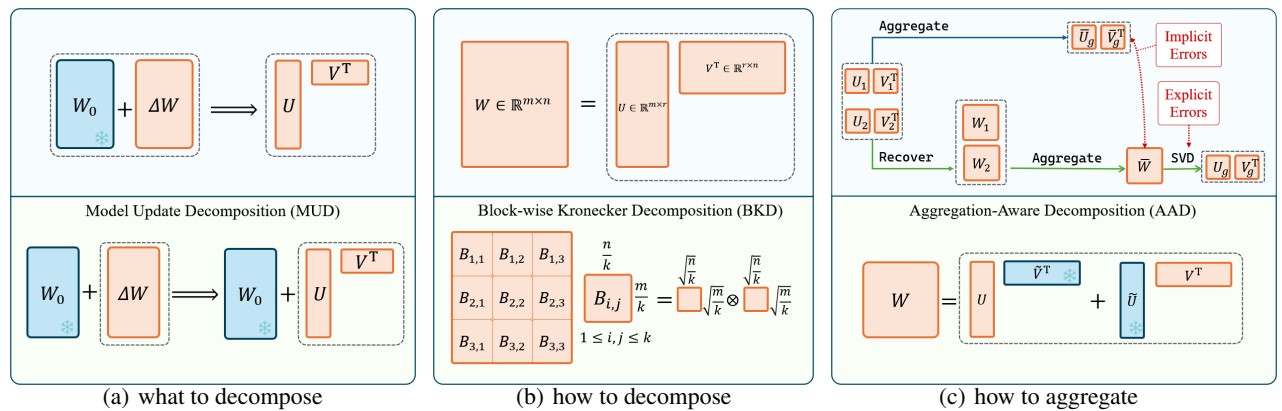

*Figure 1.* An illustration of three key issues related to decomposition in FL and their corresponding solutions: **(a) What to decompose to minimize information loss?** Existing methods decomposes the entire parameters, while MUD decomposes only the model update, effectively reducing information loss. **(b) How to decompose to achieve a higher rank?** Standard low-rank decomposition achieves $\text{rank}(W) \leq \min r \ll \min\{m, n\}$ with $(m+n)r$ parameters, while BKD achieves higher rank upper bound, i.e., $\text{rank}(W) \leq \min\{m, n\}$ with $k\sqrt{mn}$ parameters. **(c) How to aggregate to reduce compression errors?** Directly aggregating sub-matrices results in implicit errors, as $\bar{W} = {}^{U_1(V_1)^\top + U_2(V_2)^\top}/2 \neq {}^{U_1(V_1)^\top + U_1(V_2)^\top + U_2(V_1)^\top + U_2(V_2)^\top}/4 = \bar{U}(\bar{V})^\top$. AAD can effectively avoid this deviation.

issues related to decomposition in FL to further enhance communication efficiency, as shown in Figure 1.

*First, what to decompose to minimize information loss?* In FL, communication is driven by the model updates derived from local training or global aggregation. Both the server and clients can generate the latest model parameters by transmitting only the model updates. Therefore, we argue that decomposing model updates is more efficient than decomposing the entire model parameters, as the latter may lead to greater information loss. Building on this insight, we propose **Model Update Decomposition (MUD)**, which freezes the original model parameters and then learns low-rank sub-matrices to serve as model updates during local training, as illustrated in Figure 1(a).

*Second, how to decompose to achieve a higher rank?* The rank of a matrix indicates the amount of information it encodes, so maximizing the rank of the recovered matrix is crucial in matrix decomposition (Hyeon-Woo et al., 2022). To achieve this, we propose **Block-wise Kronecker Decomposition (BKD)**, which partitions a matrix into blocks and then decomposes each block with the Kronecker product, as shown in Figure 1(b). BKD offers a higher rank upper bound for the recovered matrix while requiring fewer parameters. Furthermore, the block structure of BKD enables dynamic compression by varying the number of blocks, $k$, similar to adjusting the rank, $r$, in traditional low-rank decomposition.

*Third, how to aggregate to reduce compression errors?* As shown in Figure 1(c), after receiving the optimized low-rank models, the server should aggregate the recovered matrices and then obtain a new low-rank model through SVD, which explicitly introduces errors into the model parameters.

Consequently, Liu et al. (2024) suggest directly aggregating the sub-matrices to avoid the SVD process. However, our analysis indicates that this aggregation introduces implicit errors. Specifically, we find that $\bar{U}(\bar{V})^\top \neq \bar{W}$, where $\bar{U}$, $\bar{V}$ and $\bar{W}$ are the aggregated results of the sub-matrices $U$, $V$ and the recovered matrix $W$, respectively. To address this issue, we propose **Aggregation-Aware Decomposition (AAD)**, which decouples the multiplication of trainable sub-matrices, as shown in Figure 1(c). Through AAD, there is no discrepancy between recovering then aggregating and aggregating then recovering, i.e., $\bar{W} = \bar{U}\tilde{V} + \tilde{U}\bar{V}$, where $\tilde{U}$ and $\tilde{V}$ are fixed matrices shared among clients. AAD applies to all forms of products involving trainable parameters that require aggregation, including the Kronecker product.

The contributions of this paper are summarized as follows:

- A comprehensive study on low-rank decomposition in CEFL is presented, along with three novel techniques to enhance performance: Model Update Decomposition, Block-wise Kronecker Decomposition, and Aggregation-Aware Decomposition.

- A rigorous theoretical analysis of the proposed method is provided, confirming its effectiveness and convergence. Specifically, it is demonstrated that the proposed method converges faster than existing low-rank decomposition approaches in FL, such as FedLMT.

- Extensive experiments are conducted on four popular datasets to evaluate the superiority of the proposed method. Notably, our method achieves up to a 12% improvement in test accuracy compared to relevant baselines, with each technique contributing positively.

## 2. Preliminaries

The goal of FL is to train a global model using decentralized datasets $\{\mathbb{D}_1, \mathbb{D}_2, \ldots, \mathbb{D}_N\}$, which can be formulated as:

$$\min f(\mathbf{w}) \triangleq \frac{1}{N} \sum_{i=1}^{N} f_i(\mathbf{w}), \qquad (1)$$

where $\mathbf{w}$ denotes the model parameters. $f_i(\cdot)$ denotes the expected loss function of client $i$, defined as $f_i(\mathbf{w}) \triangleq \mathbb{E}_{\xi \in \mathbb{D}_i}[F_i(\mathbf{w}, \xi)]$, where $F_i(\mathbf{w}, \xi)$ is the loss value corresponding to the data sample $\xi$. For simplicity, we will omit the symbol $\xi$ in $F_i(\cdot)$ in the following discussions.

To solve Eq.(1) in a privacy-preserving manner, FL generally requires multiple rounds of training. In each round $t$, the server sends the global model parameters $\mathbf{w}^t$ to clients. Each client $i$ then optimizes $\mathbf{w}^t$ using local data through several steps of gradient descent, yielding $\mathbf{w}_i^{t+1}$. Subsequently, each client $i$ sends the local model update, $\Delta \mathbf{w}_i^t = \mathbf{w}_i^{t+1} - \mathbf{w}^t$, back to the server, which aggregates these updates to generate the global model update as follows:

$$\Delta \mathbf{w}^t = \frac{1}{N} \sum_{i=1}^{N} \Delta \mathbf{w}_i^t. \qquad (2)$$

Using $\Delta \mathbf{w}^t$, the global model is updated as $\mathbf{w}^{t+1} = \mathbf{w}^t + \Delta \mathbf{w}^t$. In the next round, the server can send either $\mathbf{w}^{t+1}$ or just $\Delta \mathbf{w}^t$ to the clients. In the latter case, clients reconstruct $\mathbf{w}^{t+1}$ with their locally stored $\mathbf{w}^t$. Notably, even if a client does not participate in a specific round, it must still download the global model update to ensure correct restoration of the model parameters in future rounds. However, the communication overhead incurred by non-participating clients does not impact the overall training efficiency of FL.

## 3. Methodology

### 3.1. Model Update Decomposition

Recent approaches in FL reduce communication overhead by training low-rank models on the client side (Hyeon-Woo et al., 2022). For example, FedHM (Yao et al., 2021) generates low-rank models by applying truncated SVD to the global model, while FedLMT (Liu et al., 2024) directly trains a pre-decomposed global model.

However, we argue that these approaches result in substantial information loss, which consequently reduces accuracy. Taking the parameter $W$ as an example, the communication content in round $t$ can be expressed as $W^{t+1} = W^t + \Delta W^t$, where $\Delta W^t$ denotes either a local or global model update. Decomposing the entire model parameter $W^{t+1}$ can introduce significant errors in both $W^t$ and $\Delta W^t$. Since both the server and clients typically have access to $W^t$, it is sufficient and more efficient to decompose and transmit only the

model update $\Delta W^t$, as this avoids introducing errors into $W^t$. To this end, we propose **Federated Learning with Model Update Decomposition (FedMUD)**. As shown in Figure 1, FedMUD freezes the original model parameters and then learns low-rank matrices to serve as model updates. In round $t$ and on client $i$, the parameter $W$ is optimized as:

$$W_i^{t+1} = W^t + \Delta W_i^t = W^t + U_i^t (V_i^t)^\top, \qquad (3)$$

where $W^t$ represents the received global parameter, which remains frozen during local training. The trainable component is the pre-decomposed model update, denoted as $U_i^t$ and $V_i^t$. Prior to local training, $U_i^t$ is initialized randomly, while $V_i^t$ is initialized to zero, ensuring that the model update starts at zero. Additionally, the server sends a random seed to the clients to ensure consistent initialization of $U$ across all clients. After local training, the clients send their optimized $U$ and $V$ back to the server, which aggregates these sub-matrices as follows:

$$\bar{U}^t = \frac{1}{N} \sum_{i=1}^{N} U_i^t, \quad \bar{V}^t = \frac{1}{N} \sum_{i=1}^{N} V_i^t. \qquad (4)$$

In the next round, the server sends $\bar{U}_l^t$ and $\bar{V}_l^t$ back to the clients. Upon receiving $\bar{U}_l^t$ and $\bar{V}_l^t$, the clients have two options: (1) incorporate these sub-matrices into the frozen parameters and then reinitialize the sub-matrices, or (2) continue training with the received sub-matrices. Let $s$ denote the number of rounds for resetting model updates, referred to as the reset interval. Every $s$ rounds, the recovered matrix $\bar{U}(\bar{V})^\top$ is added to the frozen parameters. Assuming the total number of training rounds is $R = ns$, the final global model parameters can be expressed as follows:

$$W^R = W^0 + \sum_{\tau=1}^{n} \bar{U}^{\tau s} \bar{V}^{\tau s}, \qquad (5)$$

which implies that the initialized parameters are updated through the accumulation of low-rank updates. As $s$ decreases, the number of low-rank updates increases, thereby improving the information richness. To achieve optimal accuracy, $s$ is set to 1 by default. Notably, by setting $W^0 = \mathbf{0}$, $s \geq R$ and initializing both $U$ and $V$ randomly, FedMUD reduces to FedLMT. In other words, the parameters of FedLMT correspond solely to a low-rank update of FedMUD, causing FedLMT to perform much worse than FedMUD. Theoretically, we show that increasing $s$ can negatively affect the convergence rate of FedMUD in Section 4.

### 3.2. Block-wise Kronecker Decomposition

In the previous section, we demonstrated the necessity of decomposing the model update. With standard low-rank decomposition, the model update $\Delta W \in \mathbb{R}^{m \times n}$ is represented as the product of two smaller matrices, $U \in \mathbb{R}^{m \times r}$ and

$V \in \mathbb{R}^{n \times r}$. The compression ratio is $2(m+n)r/mn$, where $r \ll \min\{m, n\}$. However, due to the property of matrix multiplication, the rank of the recovered model update is constrained as $\mathrm{rank}(\Delta W) \leq \min\{\mathrm{rank}(U), \mathrm{rank}(V)\} \leq r$, which limits the choice of smaller values for $r$.

To address this issue, we utilize the Kronecker product to enhance the rank of the recovered matrix. Assume $\sqrt{m}$ and $\sqrt{n}$ are integers, the Kronecker decomposition can be formulated as $\Delta W = U \otimes V$, where $U, V \in \mathbb{R}^{\sqrt{m} \times \sqrt{n}}, \Delta W \in \mathbb{R}^{m \times n}$, and $\otimes$ denotes the Kronecker product. The resulting compression ratio is $2/\sqrt{mn}$, and the rank satisfies $\mathrm{rank}(\Delta W) = \mathrm{rank}(U) \times \mathrm{rank}(V) \leq \min\{m, n\}$, which matches the upper bound for a matrix of dimensions $(m, n)$.

However, the compression ratio of the Kronecker decomposition, specifically $2/\sqrt{mn}$, is overly rigid and lacks dynamic adaptability. To address this limitation, we introduce **Block-wise Kronecker Decomposition (BKD)**, which partitions the matrix into $k^2$ blocks and applies the Kronecker decomposition to each block, as shown in Figure 1. Assume $m = ka^2, n = kb^2$, a matrix $W \in \mathbb{R}^{m \times n}$ will be represented by $k^2$ pairs of sub-matrices $U, V \in \mathbb{R}^{a \times b}$. Consequently, the compression ratio is given by $2k^2 ab/mn = 2k/\sqrt{mn}$. Dynamic compression can be achieved by adjusting $k$, like adjusting $r$ in the standard low-rank decomposition. Notably, when $k = 1$, BKD reduces to the Kronecker decomposition. The rank upper bound of BKD is full rank, similar to that of the Kronecker product-based decomposition discussed above. A detailed analysis of the rank upper bound of BKD is provided in Appendix B.

Decomposition is commonly applied to linear and convolutional layers. The parameter of a linear layer is a two-dimensional tensor, making low-rank decomposition straightforward. In contrast, the parameter of a convolutional layer is a four-dimensional tensor $W \in \mathbb{R}^{c_{out} \times c_{in} \times k \times k}$, which should be reshaped into a two-dimensional tensor $W' \in \mathbb{R}^{c_{out}k \times c_{in}k}$ before decomposition. This strategy has also been utilized in (Liu et al., 2024). In BKD, we let the sub-matrices be square for simplicity, i.e., $a = b = z$, so that the restored matrix becomes $W \in \mathbb{R}^{kz^2 \times kz^2}$. Here, $k^2$ represents the number of blocks, and $z$ is determined by $k$ and the size of the target tensor. For instance, in the case of a linear layer with dimensions $(m, n)$, $z$ is computed as $z = \lceil \sqrt[4]{mn/k^2} \rceil$. When $k^2 z^4 > mn$, only the first $mn$ parameters of $W \in \mathbb{R}^{kz^2 \times kz^2}$ are reshaped into the target parameter matrix with dimensions $(m, n)$. This process is applicable to tensors of any dimension.

### 3.3. Aggregation-Aware Decomposition

Aggregation in FL is to synthesize the knowledge learned by different clients. This is commonly achieved by averaging the parameters of different clients or through weighted averaging based on the data volume each client holds. However, low-rank decomposition presents new challenges for parameter aggregation. Specifically, there are two ways to aggregate the sub-matrices: direct aggregation and aggregation after recovery. In direct aggregation, the results are also sub-matrices, which can be directly sent to clients for the next training round. In contrast, aggregation after recovery requires applying truncated SVD on the aggregated results to obtain new sub-matrices before transmission to the clients. However, the SVD process introduces approximation errors, which can significantly affect convergence. Therefore, we primarily focus on the direct aggregation.

However, directly aggregating the sub-matrices, $U$ and $V$, introduces bias into the model parameters implicitly. This issue persists regardless of whether $U$ and $V$ are used to represent the model parameters or the model update. For simplicity, we will explain this bias in the context of the former scenario. Let $U_0$ and $V_0$ denote the initial parameters received by the clients, while $U_i = U_0 + \Delta U_i$ and $V_i = V_0 + \Delta V_i$ denote the optimized parameters on client $i$. The recovered matrix, which represents the parameters actually used during the client's forward propagation, is given by $W_i = U_i(V_i)^\top$. Their aggregated result is:

$$W^* = \mathcal{A}[U_i(V_i)^\top] = U_0(V_0)^\top + U_0\mathcal{A}(\Delta V)^\top + \mathcal{A}(\Delta U)(V_0)^\top + \mathcal{A}(\Delta U(\Delta V)^\top), \tag{6}$$

where $\mathcal{A}(x)$ returns the aggregated result of variable $x$. In direct aggregation, the aggregated results are $\bar{U} = U_0 + \mathcal{A}(\Delta U)$ and $\bar{V} = V_0 + \mathcal{A}(\Delta V)$. The recovered matrix of $\bar{U}$ and $\bar{V}$ is then given by:

$$W = \bar{U}(\bar{V})^\top = U_0(V_0)^\top + U_0\mathcal{A}(\Delta V)^\top + \mathcal{A}(\Delta U)(V_0)^\top + \mathcal{A}(\Delta U)\mathcal{A}(\Delta V)^\top. \tag{7}$$

It is evident that $W$ in Eq.(7) deviates from the desired result $W^*$ in Eq.(6). The discrepancy arises from the last term, i.e., the second-order update induced by $\Delta U$ and $\Delta V$. To address this issue, we propose decoupling the multiplication of the trainable matrices $U$ and $V$ to eliminate the second-order update. Specifically, we introduce **Aggregation-Aware Decomposition (AAD)** , which reformulates the operation of $U(V)^\top$ as follows:

$$W = U(\tilde{V})^\top + \tilde{U}(V)^\top, \tag{8}$$

where $\tilde{U}$ and $\tilde{V}$ are randomly initialized and remain fixed during local training. Through AAD, the actual and desired recovered matrices are equal as follows:

$$W = W^* = (U_0 + \mathcal{A}(\Delta U))(\tilde{V})^\top + \tilde{U}(V_0 + \mathcal{A}(\Delta V))^\top. \tag{9}$$

When AAD is applied to model updates, both $U$ and $V$ shall be initialized to zeros, ensuring that the model update

starts at zero. It is important to note that AAD is applicable to any product involving trainable parameters that require aggregation, including matrix multiplication and the Kronecker product. Therefore, AAD can also be used to enhance BKD. Integrating AAD with BKD requires only replacing the matrix multiplication in Eq.(8) with the proposed BKD operator.

## 4. Convergence Analysis

In this section, we present a theoretical analysis that demonstrates the convergence of FedMUD and highlights its advantages over FedLMT. Our objective is to compare the convergence rates of full-weight decomposition and model update decomposition (i.e., FedLMT vs. FedMUD); therefore, BKD and AAD are not considered in this analysis. The notations used are summarized below.

**Notations.** For an $L$-layer neural network, the model parameters are represented as $\mathbf{w} = \{W_1, \ldots, W_\rho, W_{\rho+1} + U_{\rho+1}(V_{\rho+1})^\top, \ldots, W_L + U_L(V_L)\top\}$, where MUD is applied to the last $(L - \rho)$ layers. $R$ denotes the number of training rounds, $E$ denotes the number of iterations per round, and $T = RE$ denotes the total number of iterations. $\|\cdot\|_2$ denotes the $\ell_2$ norm of a vector and $\|\cdot\|_F$ denotes the Frobenius norm of a matrix. In this section, $t \in [T]$ denotes the current iteration number, rather than the training round. For the sake of analysis, we make the following assumptions, as used by FedLMT (Liu et al., 2024).

**Assumption 1.** *Each loss function $f_i$ (for $i = 1, 2, \ldots, N$) is differentiable and $L_s$-smooth.*

**Assumption 2.** *The stochastic gradient $\nabla F_i(\mathbf{w})$ is unbiased, with bounded variance and norm. Specifically, $\mathbb{E}\nabla F_i(\mathbf{w}) = \nabla f_i(\mathbf{w})$, $\mathbb{E}\|\nabla F_i(\mathbf{w}) - \nabla f_i(\mathbf{w})\|_2^2 \leq \sigma^2$, and $\mathbb{E}\|\nabla F_{i,l}(\mathbf{w})\|_F^2 \leq G^2$.*

**Assumption 3.** *The sub-matrices $U$ and $V$ are bounded during training, i.e., $\|U_l\|_F^2 \leq \kappa_u^2$ and $\|V_l\|_F^2 \leq \kappa_v^2$. Furthermore, the initialization values of $U$ and $V$ also satisfy $\|U_l\|_F^2 \leq \epsilon_u^2 \ll \kappa_u^2$ and $\|U_l\|_F^2 \leq \epsilon_v^2 \ll \kappa_v^2$.*

**Assumption 4.** *At least one of the matrices $\bar{U}_l = \frac{1}{N}\sum_{i=1}^N U_{i,l}$ and $\bar{V}_l = \frac{1}{N}\sum_{i=1}^N V_{i,l}$ has a positive smallest singular value, i.e., $[\delta_{min}(\bar{U}_l)]^2 + [\delta_{min}(\bar{V}_l)]^2 \geq \psi_{uv}^2 > 0$, where $\delta_{min}(\cdot)$ denotes the smallest singular value.*

Assumptions 1 and 2 are commonly used in convergence analysis under distributed settings (Li et al., 2020; Yu et al., 2019; Li et al., 2024b). Assumption 3 is also commonplace in the theoretical analysis of machine learning models (Chen et al., 2020; Liu et al., 2024). In our case, this assumption holds exactly; see Appendix C.2 for details. Given that the model dimension is finite, it is reasonable to assume that the model weights are bounded. Assumption 4 is supported by the Marchenko-Pastur theory, as discussed by Liu et al. (2024). Based on these assumptions, we present Theorem 1,

which demonstrates the convergence of FedMUD with non-convex settings. The proof is provided in Appendix C.

**Theorem 1.** *Under Assumptions 1, 2, 3 and 4, let $1 < c < 2$ be a constant and the learning rate satisfy $0 < \eta \leq \min\{(\frac{\psi_{uv}^2}{2})^{\frac{1}{c-1}}, \frac{1}{L_s}, 1\}$, with FedMUD, we have*

$$\frac{1}{T}\sum_{t=1}^T \mathbb{E}\left\|\nabla f(\mathbf{w}^{t-1})\right\|_2^2 \leq \frac{2}{\eta^c T}(f(\mathbf{w}^0) - f(\mathbf{w}^*))$$
$$+ \mathcal{O}(\eta^{2-c})\left[1 + (L - \rho)\mathcal{O}(\Gamma_u^4 + \Gamma_v^4)\right] \tag{10}$$

*where $\mathbf{w}^*$ denotes the optimal parameters, $\Gamma_u^2 \triangleq \min\{2\epsilon_u^2 + 2\eta^2 S^2 G^2 \kappa_u^2, \kappa_u^2\}$ and $\Gamma_v^2 \triangleq \min\{2\epsilon_v^2 + 2\eta^2 S^2 G^2 \kappa_u^2, \kappa_v^2\}$.*

**Remark 1.** *By setting $\rho = L$, Theorem 1 is equivalent to the convergence analysis of FedAvg. Additionally, by setting $\eta = \frac{1}{\sqrt{T}}$ and letting $c \to 1$, the convergence rates of both FedAvg and FedMUD are $\mathcal{O}(\frac{1}{\sqrt{T}})$.*

**Remark 2.** *Compared to FedAvg, the convergence rate of FedMUD is further affected by the term $(L - \rho)\mathcal{O}(\Gamma_u^4 + \Gamma_v^4)$. As $S \to T$, $\Gamma_u$ and $\Gamma_v$ gradually increase, eventually reaching their respective upper bounds, $\kappa_u$ and $\kappa_v$. Thus, as $S$ increases, the convergence of FedMUD becomes slower.*

**Remark 3.** *By setting $S \geq T$, $W_l = \mathbf{0}$ and initializing both $U_l$ and $V_l$ randomly for all $l > \rho$, Theorem 1 reduces to the convergence of FedLMT. As noted in Remark 2, FedLMT employs a much larger value for $S$, which results in a slower convergence rate compared to FedMUD.*

## 5. Experiments

### 5.1. Experimental Setup

**Datasets and Models.** In this section, we evaluate the proposed method on four widely used datasets: FMNIST (Xiao et al., 2017), SVHN (Netzer et al., 2011), CIFAR-10, and CIFAR-100 (Krizhevsky & Hinton, 2009). For FMNIST and SVHN, we employ a convolutional neural network (CNN) with four convolutional layers and one fully connected layer. For CIFAR-10 and CIFAR-100, we employ a CNN with eight convolutional layers and one fully connected layer. ReLU (Glorot et al., 2011) is used as the activation function, and batch normalization (BN) (Ioffe & Szegedy, 2015) is utilized to ensure stable training. For better performance, we do not compress the first and last layers of the model.

**Data Partitioning.** Each dataset is partitioned into several subsets to serve as the local data of different clients. Based on the data partitioning benchmark of FL (Li et al., 2022), we consider two kinds of non-IID data distribution, termed Non-IID-1 and Non-IID-2. In Non-IID-1, the proportion of the same label across clients follows the Dirichlet distribution (Yurochkin et al., 2019), while in Non-IID-2, each client only contains data of partial labels. For CIFAR-100,

*Table 1.* Accuracy of different methods. Except for FedAvg, the best accuracy is in bold and the second best accuracy is underlined.

| | FMNIST | | SVHN | | CIFAR-10 | | CIFAR-100 | |
|---|---|---|---|---|---|---|---|---|
| | Non-IID-1 | Non-IID-2 | Non-IID-1 | Non-IID-2 | Non-IID-1 | Non-IID-2 | Non-IID-1 | Non-IID-2 |
| FedAvg (McMahan et al., 2017) | 90.3($\pm$ 0.2) | 88.6 ($\pm$ 0.3) | 89.7($\pm$ 0.3) | 88.3($\pm$ 0.3) | 81.0($\pm$ 0.7) | 77.8($\pm$ 0.7) | 49.8($\pm$ 0.6) | 44.8($\pm$ 0.5) |
| FedHM (Yao et al., 2021) | 87.2($\pm$ 0.1) | 84.5($\pm$ 0.4) | 73.8($\pm$ 0.8) | 71.1($\pm$ 0.9) | 66.5($\pm$ 0.4) | 63.3($\pm$ 0.7) | 28.5($\pm$ 0.8) | 23.3($\pm$ 0.3) |
| FedLMT (Liu et al., 2024) | 87.3($\pm$ 0.4) | 84.4($\pm$ 0.5) | 71.0($\pm$ 0.6) | 70.1($\pm$ 1.0) | 64.7($\pm$ 0.8) | 62.1($\pm$ 0.8) | 28.4($\pm$ 0.3) | 23.4($\pm$ 0.6) |
| FedPara (Hyeon-Woo et al., 2022) | 87.1($\pm$ 0.2) | 84.4($\pm$ 0.3) | 82.4($\pm$ 0.6) | 79.3($\pm$ 0.5) | 70.6($\pm$ 0.6) | 66.6($\pm$ 0.7) | 29.3($\pm$ 0.9) | 25.3($\pm$ 0.6) |
| EF21-P (Gruntkowska et al., 2023) | 84.1($\pm$ 0.8) | 82.0($\pm$ 0.8) | 82.5($\pm$ 0.5) | 76.7($\pm$ 0.9) | 47.2($\pm$ 0.8) | 46.8($\pm$ 0.9) | 15.2($\pm$ 0.7) | 13.8($\pm$ 1.0) |
| FedBAT (Li et al., 2024b) | 88.4($\pm$ 0.1) | 86.7($\pm$ 0.1) | 85.2($\pm$ 0.4) | 83.8($\pm$ 0.3) | 75.0($\pm$ 0.1) | 72.3($\pm$ 0.4) | 37.4($\pm$ 0.7) | 34.2($\pm$ 0.6) |
| FedMUD | 87.9($\pm$ 0.2) | 85.8($\pm$ 0.4) | 81.3($\pm$ 0.8) | 78.5($\pm$ 0.7) | 68.9($\pm$ 0.9) | 65.8($\pm$ 0.4) | 29.4($\pm$ 0.5) | 28.9($\pm$ 0.4) |
| FedMUD+BKD | 88.2($\pm$ 0.1) | 86.7($\pm$ 0.6) | 84.8($\pm$ 0.5) | 83.5($\pm$ 0.4) | 70.2($\pm$ 0.7) | 67.5($\pm$ 0.8) | 35.7($\pm$ 0.5) | 30.7($\pm$ 0.8) |
| FedMUD+AAD | 88.5($\pm$ 0.3) | 87.0($\pm$ 0.3) | 84.7($\pm$ 0.2) | 82.5($\pm$ 0.6) | 73.3($\pm$ 0.4) | 72.3($\pm$ 0.3) | 37.4($\pm$ 0.4) | 35.1($\pm$ 0.2) |
| FedMUD+BKD+AAD | **89.0($\pm$ 0.2)** | **87.6($\pm$ 0.2)** | **86.6($\pm$ 0.3)** | **84.9($\pm$ 0.3)** | **75.9($\pm$ 0.2)** | **73.9($\pm$ 0.6)** | **41.2($\pm$ 0.5)** | **36.1($\pm$ 0.9)** |

we set the Dirichlet parameter to 0.1 in Non-IID-1 and assign 10 random labels to each client in Non-IID-2. For the other datasets, we set the Dirichlet parameter to 0.3 in Non-IID-1 and assign 3 random labels to each client in Non-IID-2. The data heterogeneity of Non-IID-2 is generally higher than that of Non-IID-1. The experiments under the IID data distribution can be found in Appendix A.

**Baselines.** FedMUD is compared to several CEFL methods, including FedHM (Yao et al., 2021), FedLMT (Liu et al., 2024), FedPara (Hyeon-Woo et al., 2022), EF21-P (Gruntkowska et al., 2023) and FedBAT (Li et al., 2024b). FedHM decomposes the global model on the server through truncated SVD and sends the decomposed model to clients for local training. FedLMT directly trains a pre-decomposed global model. FedPara enhances the rank of the recovered matrix by applying the Hadamard product. EF21-P is a state-of-the-art algorithm for communication compression with the error feedback mechanism in FL. Following the original setting of EF21-P, the Rand-$K$ and Top-$K$ compressors are used to compress the local and global model updates, respectively. FedBAT is an advanced communication quantization algorithm in FL, which learns binary model updates during local training. However, it only compresses uplink communication. For a fair comparison, we also use its quantizer to compress the global model update.

**Hyperparameters.** The number of clients is set to 100, with 10 clients randomly selected to participate in each round of training. The local epoch is set to 3, and the batch size is 64. SGD (Bottou, 2010) is employed as the local optimizer, with the learning rate tuned from the set $\{1.0, 0.3, 0.1, 0.03, 0.01\}$. The number of training rounds is set to 100 for FMNIST and SVHN, and 200 for CIFAR-10 and CIFAR-100. The initialization of sub-matrices in the decomposition methods has a significant impact on model performance. To ensure a fair comparison, we adjust the initialization across methods to optimize performance. Specifi-

cally, the sub-matrices are randomly initialized with values drawn from the uniform distribution $U(-a, a)$, where $a$ is selected from the set $\{0.01, 0.05, 0.1, 0.5, 1, 5, 10\}$. In the main experiments, the compression ratio is set to $1/32$ for all methods. For FedMUD, we also report the performance of its variants, corresponding to whether BKD or AAD is applied on top of MUD. Each experiment is run five times, and the average results are reported. The code is provided in the supplementary materials.

## 5.2. Overall Performance

In this section, we compare the performance of our methods and the baselines in terms of test accuracy and convergence speed. All numerical results are reported in Table 1, and partial convergence curves are shown in Figure 2. More experimental results can be found in Appendix A.

First, we observe that FedHM and FedLMT exhibit similar performance. As discussed in Section 1, these two methods introduce errors into model parameters in distinct manners. Specifically, the errors in FedHM stem from truncated SVD, while those in FedLMT arise from biases introduced by directly aggregating sub-matrices. In contrast, FedPara achieves higher accuracy by increasing the rank of the recovered matrix, which highlights the crucial role of matrix rank in low-rank decomposition methods. Compared to these decomposition methods, FedMUD consistently demonstrates superior accuracy and faster convergence. Moreover, its performance is further enhanced when integrated with the BKD and AAD techniques. For EF21-P, we attribute its poor performance to the excessively high compression rate, with only 3% of the parameters being updated per round. Among the baselines, FedBAT performs the best, notably surpassing existing low-rank decomposition methods. However, our method still exceeds FedBAT by a substantial margin, bridging the gap in low-rank decomposition for CEFL.

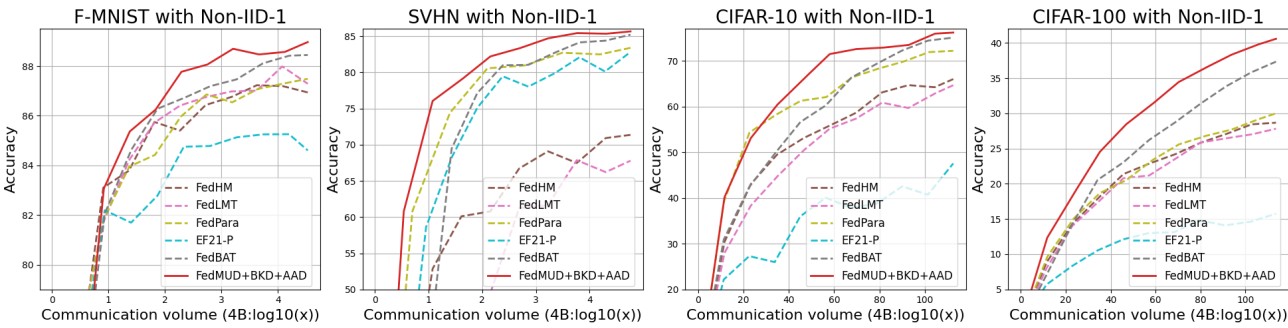

*Figure 2.* Convergence curves of different methods under the Non-IID-1 data distribution.

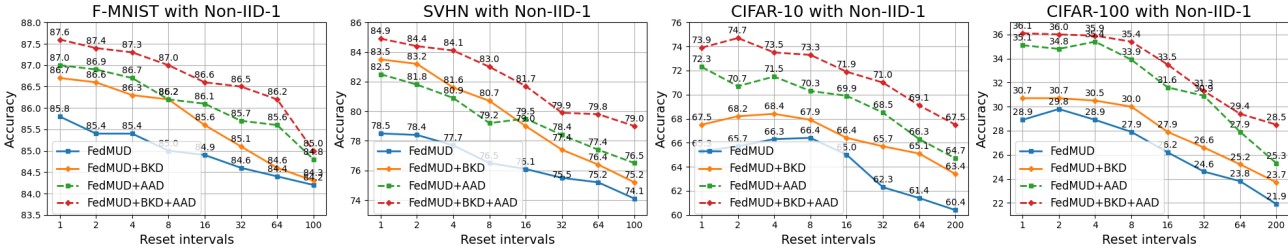

*Figure 3.* Accuracy of FedMUD under different reset intervals.

*Table 2.* Accuracy of FedMUD when applied with freezing (+F) and decoupling (+AAD) under the Non-IID-1 data distribution.

|  | FMNIST | SVHN | CIFAR-10 | CIFAR-100 |
|---|---|---|---|---|
| FedMUD+F | **88.5** | 81.0 | 71.1 | 35.3 |
| FedMUD+AAD | **88.5** | **84.7** | **73.3** | **37.4** |
| FedMUD+BKD+F | 88.6 | 84.9 | 74.4 | 38.0 |
| FedMUD+BKD+AAD | **89.0** | **86.6** | **75.9** | **41.2** |

### 5.3. Decoupling vs. Freezing

ADD decouples the multiplication of trainable matrices to avoid errors introduced by aggregating sub-matrices. In principle, a similar effect could also be achieved by freezing one of the sub-matrices (Sun et al., 2024). Therefore, we conduct an ablation experiment to compare the difference between decoupling and freezing, under the same communication cost. Table 2 demonstrates that AAD consistently outperforms the method that freezes the matrix $U$, where FedMUD+AAD and FedMUD+F utilize different update matrix shapes: $U_{AAD}(\tilde{V})^\top + \tilde{U}(V_{AAD})^\top$ for the former and $\tilde{U}(V_F)^\top$ for the latter. Although $U_{AAD}$ and $V_{AAD}$ individually have significantly fewer parameters than $V_F$, their sum is approximately equal. As discussed by Hayou et al. (2024), the roles of $U$ and $V$ in low-rank training differ, and both of their updates significantly affect convergence. Therefore, freezing either $U$ or $V$ results in suboptimal performance. In contrast, AAD can be viewed as an additive combination of two freezing modules, which ensures the simultaneous trainability of $U$ and $V$.

### 5.4. Ablation on Reset Intervals

By default, FedMUD reinitializes the sub-matrices before each round of local training, i.e., the reset interval $s = 1$. In this section, we examine the effect of the reset interval on model performance. As illustrated in Figure 3, model accuracy generally decreases as the reset interval increases, consistent with our theoretical analysis. However, compared to $s = 1$, a slight improvement in model accuracy is occasionally observed for $s = 2$ or $s = 4$, which can be attributed to data heterogeneity. Specifically, a larger interval allows the low-rank matrices to observe more diverse data before being added into the frozen parameters, thereby mitigating the effects of data heterogeneity. Yet, as the interval continues to increase, previously learned knowledge may be forgotten or overwritten, leading to a decline in accuracy. Moreover, when the reset interval equals the total number of training rounds, FedMUD reduces to FedLMT, and its performance becomes comparable to that of FedLMT. Figure 3 also demonstrates the performance improvements achieved by AAD and BKD individually, with AAD showing a more significant contribution. In conclusion, the reset interval should typically be set to 1 for optimal performance.

### 5.5. Ablation on Initialization Values

The initialization of sub-matrices plays a critical role in the performance of low-rank decomposition methods. In our experiments, we employed a uniform distribution $U(-a, a)$ to initialize the sub-matrices. This section examines how different initialization values affect the performance of FedMUD.

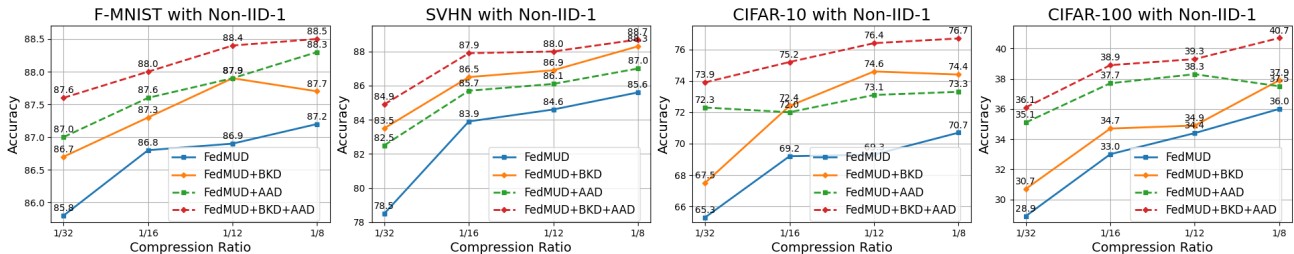

*Figure 4.* Accuracy of FedMUD under different initialization values.

*Figure 5.* Accuracy of FedMUD under different compression ratios.

In Figure 4, the $x$-axis denotes the parameter $a$, which determines the magnitude of the initialization range. As illustrated, we observe that inappropriate initialization values can lead to a decrease in model accuracy. In contrast, the proposed BKD exhibits reduced sensitivity to variations in initialization values. Further, BKD typically requires larger initialization values compared to traditional low-rank decomposition. Overall, some adjustment of the initialization values is necessary for FedMUD to achieve optimal performance. Based on these observations, we believe that it is feasible to dynamically adjust the initialization values for FedMUD during training, and we leave this as future work.

### 5.6. Ablation on Compression Ratios

The compression ratio is set to $1/32$ for all methods in Section 5.2, where FedMUD significantly outperforms the baselines in terms of accuracy, though it still lags behind FedAvg. This section examines the impact of varying compression ratios on model performance. As illustrated in Figure 5, the accuracy improves as the compression strength decreases. Notably, with 8-fold compression, FedMUD+BKD+AAD achieves accuracy comparable to that of FedAvg.

## 6. Related Work

### 6.1. Low-Rank Decomposition

Low-rank decomposition is a matrix compression technique that expresses a matrix as the product of two sub-matrices (Sainath et al., 2013). To enhance performance, researchers have further explored decomposing matrices into more sub-matrices, as in Tucker Decomposition (Malik & Becker,

2018) and Tensor-Train Decomposition (Novikov et al., 2018). Recently, such techniques have been widely adopted in model compression (Yao et al., 2021; Novikov et al., 2018), parameter-efficient fine-tuning (Hu et al., 2022; Mao et al., 2024), and related domains.

This paper investigates the application of low-rank decomposition in communication-efficient federated learning (CEFL). The work in (Konečný et al., 2016) was the first to apply low-rank decomposition for reducing communication costs, although it only compresses the uplink communication. As previously discussed, subsequent studies (Yao et al., 2021; Liu et al., 2024) face several challenges. To address these, we propose three novel techniques that significantly improve the performance of low-rank decomposition in federated learning. It is also noteworthy that low-rank decomposition has seen broad application in personalized federated learning (FL) (Hyeon-Woo et al., 2022; Liu et al., 2024; Wu et al., 2024; Dadras et al., 2024), and our proposed methods can be effectively extended to this setting to further enhance performance.

In addition, low-rank techniques have been extensively utilized for the efficient fine-tuning of large language models (LLMs). For example, Hu et al. (2022) introduced Low-Rank Adaptation (LoRA), which approximates updates to pretrained weights using products of low-rank matrices, significantly reducing memory consumption. However, the constrained rank may limit model performance. To overcome this, Lialin et al. (2024) proposed superimposing multiple low-rank matrices to increase the effective rank. Furthermore, Edalati et al. (2023) replaced matrix multiplication in LoRA with the Kronecker product, enabling theoreti-

cal reconstruction of full-rank matrices. Nevertheless, this approach is less flexible in controlling compression rates compared to the proposed BKD method. BoRA (Li et al., 2025b) employs a similar block-wise mechanism in LoRA, where block matrix multiplication with block-wise diagonal matrices is used to enhance the expressivity of LoRA. Li et al. (2025a) have also explored the initialization of low-rank matrices, and demonstrated the potential benefits of non-zero initialization.

### 6.2. Communication-Efficient Federated Learning

Existing methods primarily reduce communication cost in FL from two aspects: model compression and gradient compression. Model compression typically involves training a smaller model on the client side. For example, (Caldas et al., 2018; Bouacida et al., 2021) allow clients to train submodels derived from a larger server model, while (Isik et al., 2023; Bibikar et al., 2022) train pruned models during local training. Furthermore, Yang et al. (2021) trains and communicates a binary neural network. Low-rank decomposition methods, such as FedLMT and FedHM, also fall within the model compression category. However, model compression often sacrifices the model's expressiveness, resulting in decreased accuracy, especially when higher compression strengths are applied. In contrast, gradient compression targets model updates (i.e., accumulated gradients), minimizing the negative impact on model's expressiveness. Existing methods typically apply pruning (Stripelis et al., 2022; Qiu et al., 2022; Li et al., 2024a) or quantization (Karimireddy et al., 2019; Li et al., 2024b) techniques to the model updates to reduce communication overhead. Notably, MUD is a form of gradient compression. By representing the model update as sub-matrices, our approach achieves significantly higher accuracy compared to existing low-rank decomposition methods. Additionally, experimental results demonstrate that our method outperforms the gradient compression methods based on pruning and quantization.

## 7. Conclusion

In this paper, we provide a comprehensive study on low-rank decomposition techniques in the context of communication-efficient federated learning (CEFL). Specifically, we focus on three key challenges related to decomposition in FL: what to decompose, how to decompose, and how to aggregate. In response to these challenges, we propose three novel techniques: Model Update Decomposition, Block-wise Kronecker Decomposition, and Aggregation-Aware Decomposition. These techniques have been theoretically analyzed and empirically validated, and the results demonstrate that our approach significantly improves the performance of low-rank decomposition in CEFL.

## Acknowledgements

This work is supported by the National Key Research and Development Program of China under grant 2024YFC3307900; the National Natural Science Foundation of China under grants 62376103, 62302184, 62436003, and 62206102; Major Science and Technology Project of Hubei Province under grant 2024BAA008; Hubei Science and Technology Talent Service Project under grant 2024DJC078; and Ant Group through CCF-Ant Research Fund. The computation is completed in the HPC Platform of Huazhong University of Science and Technology.

## Impact Statement

In this paper, we improve low-rank decomposition techniques in communication-efficient federated learning. The proposed method focuses solely on improving the communication efficiency. Therefore, we believe that our method does not result in any adverse social impact.

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

# A. Additional Experimental Results

In this section, we present the convergence curves of different methods under various data distributions, as illustrated in Figure 6. Then, we evaluate the performance of FedMUD under the Non-IID-2 data distribution, considering various reset intervals, initialization values, and compression ratios. The experimental results, presented in Figure 7, align with the findings in Section 5. Table 4 summarizes the results of ablation experiments on decoupling and freezing, as discussed in Section 5.3, under Non-IID-2 and IID data distributions. Additionally, we provide the experimental results for all methods under the IID data distribution, as shown in Table 3. Moreover, we evaluate our method against FedAvg and FedLMT using ResNet18 (He et al., 2016) on the CIFAR-10 and TinyImageNet (Le & Yang, 2015) datasets under the Non-IID-1 setting. The compression ratio is set to $16\times$ or $32\times$, and the learning rate is tuned over $\{0.001, 0.003, 0.01, 0.03, 0.1\}$. All other experimental settings follow those described in Section 5.1. The results, summarized in Table 5, show that our method retains superior performance even with $32\times$ communication compression, achieving accuracy closest to FedAvg.

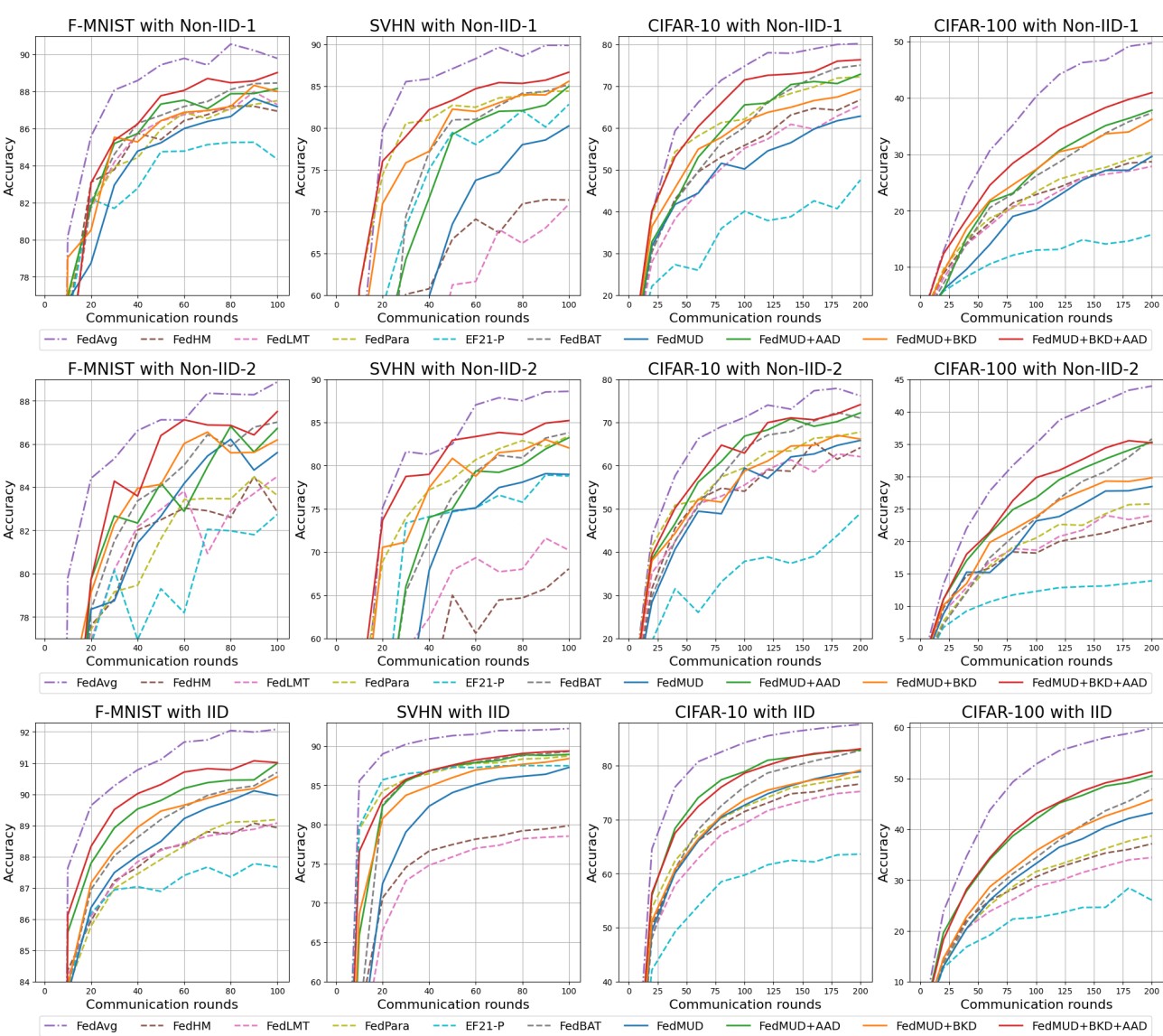

Figure 6. Convergence curves of different methods under Non-IID-1, Non-IID-2 and IID data distribution.

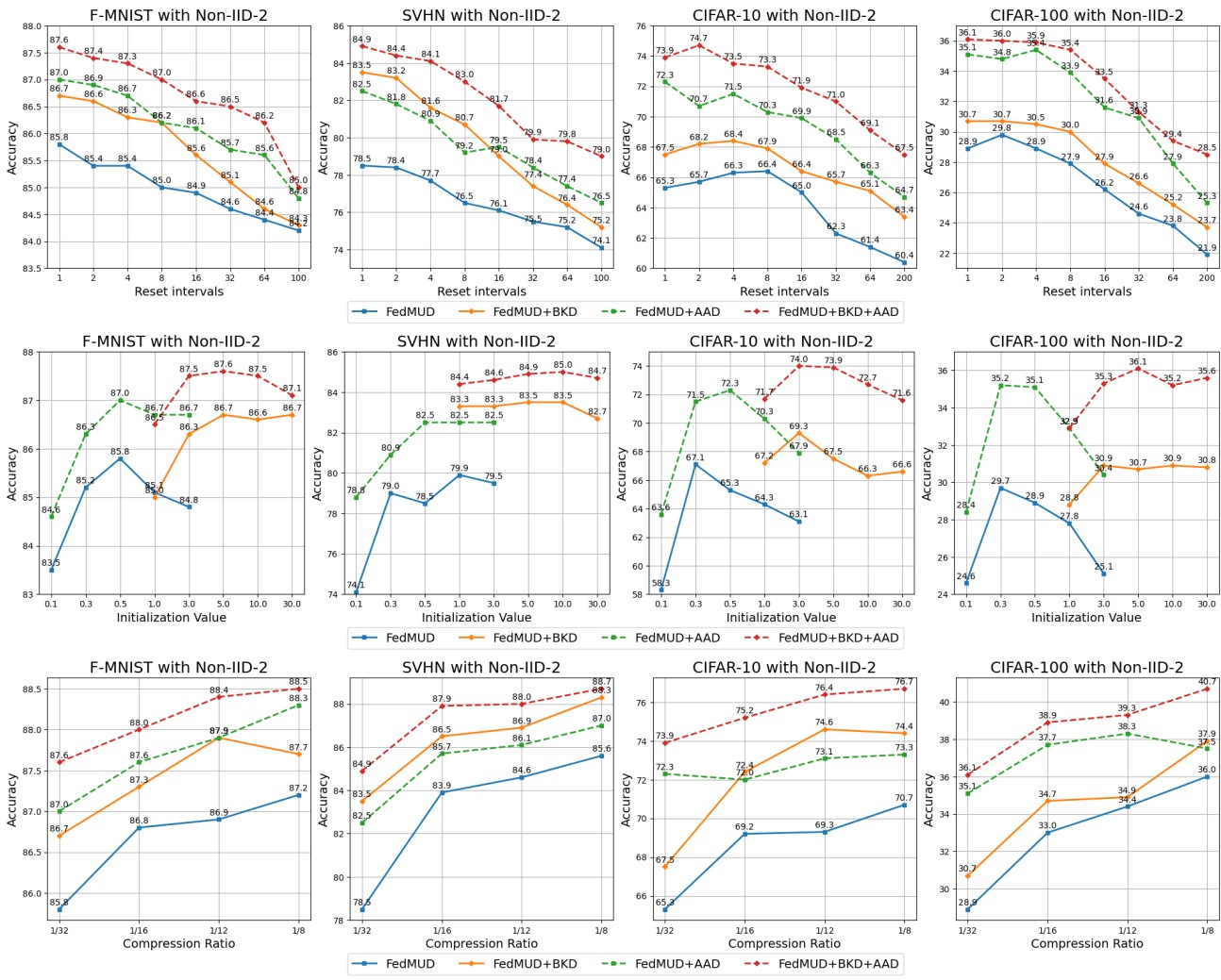

*Figure 7.* Ablation on hyperparameters for the proposed methods under Non-IID-2 and IID data distribution.

*Table 3.* Accuracy of different methods under the IID data distribution.

|  | FMNIST (IID) | SVHN (IID) | CIFAR-10 (IID) | CIFAR-100 (IID) |
|---|---|---|---|---|
| FedAvg (McMahan et al., 2017) | 91.9(± 0.1) | 92.3(± 0.1) | 87.4(± 0.2) | 59.1(± 0.4) |
| FedHM (Yao et al., 2021) | 89.1(± 0.3) | 79.9(± 0.9) | 76.4(± 0.7) | 35.9(± 0.8) |
| FedLMT (Liu et al., 2024) | 89.1(± 0.1) | 79.6(± 0.7) | 75.9(± 0.7) | 34.6(± 0.3) |
| FedPara (Hyeon-Woo et al., 2022) | 89.2(± 0.2) | 86.1(± 0.2) | 76.7(± 0.6) | 37.8(± 0.5) |
| EF21-P (Gruntkowska et al., 2023) | 87.9(± 0.3) | 87.9(± 0.2) | 64.4(± 0.4) | 28.1(± 04.) |
| FedBAT (Li et al., 2024b) | 90.5(± 0.1) | 89.3(± 0.0) | 82.8(± 0.1) | 47.3(± 0.5) |
| FedMUD | 90.0(± 0.2) | 87.3(± 0.1) | 79.2(± 0.3) | 43.5(± 0.3) |
| FedMUD+AAD | 90.9(± 0.1) | 89.1(± 0.2) | 83.0(± 0.1) | 50.5(± 0.1) |
| FedMUD+BKD | 90.5(± 0.2) | 88.5(± 0.1) | 79.0(± 0.2) | 45.7(± 0.4) |
| FedMUD+BKD+AAD | **91.0(± 0.3)** | **89.5(± 0.1)** | **83.2(± 0.3)** | **51.4(± 0.3)** |

*Table 4.* Accuracy of FedMUD when applied with freezing (+F) and decoupling (+ADD) under the Non-IID-2 and IID data distribution.

| | FMNIST | | SVHN | | CIFAR-10 | | CIFAR-100 | |
| --- | --- | --- | --- | --- | --- | --- | --- | --- |
| | Non-IID-2 | IID | Non-IID-2 | IID | Non-IID-2 | IID | Non-IID-2 | IID |
| FedMUD+F | **87.1** | **90.9** | 80.2 | 87.4 | 67.5 | 82.2 | 33.0 | 48.4 |
| FedMUD+AAD | 87.0 | **90.9** | **82.5** | **89.1** | **72.3** | **83.0** | **35.1** | **50.5** |
| FedMUD+BKD+F | 87.4 | 90.9 | 83.3 | 88.5 | 71.2 | 81.9 | 33.3 | 48.7 |
| FedMUD+BKD+AAD | **87.6** | **91.0** | **84.9** | **89.5** | **73.9** | **83.2** | **36.1** | **51.4** |

*Table 5.* Accuracy of ResNet18 on CIFAR-10 and TinyImageNet.

| | | 16x Compression | | 32x Compression | |
| --- | --- | --- | --- | --- | --- |
| Model / Dataset | FedAvg | FedLMT | FedMUD+BKD+AAD | FedLMT | FedMUD+BKD+AAD |
| ResNet18 on CIFAR-10 | 84.1 | 76.4 | **83.2** | 73.9 | **79.6** |
| ResNet18 on TinyImageNet | 40.0 | 31.3 | **36.9** | 13.8 | **33.4** |

## B. Discussion about the Upper Rank Bound of BKD

In this section, we briefly examine the upper bound on the rank of BKD by leveraging properties of the Kronecker product and matrix concatenation. Specifically, we use the identity $rank(A \otimes B) = rank(A) \cdot rank(B)$ and the inequality $rank([A|B]) \leq rank(A) + rank(B)$. Let $A, B \in \mathbb{R}^{a \times b}$; then their Kronecker product yields a matrix $W \in \mathbb{R}^{a^2 \times b^2}$. According to the rank property of the Kronecker product, we have $rank(W) = rank(A) \cdot rank(B)$, which can be as large as $\min a^2, b^2$ (i.e., the maximum possible rank for a matrix of size $a^2 \times b^2$). Since BKD can be interpreted as the concatenation of several such Kronecker product matrices, and each sub-matrix can individually attain full rank, the overall recovery matrix of BKD is capable of achieving full rank.

## C. Proof of Theorem 1

In this section, we first present the problem formulation, then outline the assumptions and lemmas necessary for the proof, and conclude with a detailed proof process. Our theoretical analysis relies mainly on FedLMT (Liu et al., 2024).

### C.1. Problem Formulation

Let us consider $N$ clients with local datasets $\mathbb{D} = \{\mathbb{D}_1, \mathbb{D}_2, \ldots, \mathbb{D}_N\}$. The goal of FL is to solve the following problem

$$\min f(\mathbf{w}) \triangleq \frac{1}{N} \sum_{i=1}^{N} f_i(\mathbf{w}), \tag{11}$$

where $\mathbf{w} = \{W_1, W_2, \ldots, W_L\}$ denotes the parameters of $L$ layers. $f_i(\mathbf{w}) \triangleq \mathbb{E}_{\xi_i \in \mathbb{D}_i}[F_i(\mathbf{w}, \xi_i)]$ is the expected loss function of client $i$ and $\xi_i$ is a random data sample of client $i$. In this paper, we propose freezing the original model parameters and learning low-rank matrices as the model updates. Assuming that the first $\rho$ layers remain unchanged and low-rank model updates are only applied to the subsequent layers, the entire set of model parameters can be expressed as

$$\mathbf{x} = \{W_1, \ldots, W_\rho, W_{\rho+1}, U_{\rho+1}, V_{\rho+1}, \ldots, W_L, U_L, V_L\}, \tag{12}$$

where $\{W_{\rho+1}, W_{\rho+2}, \ldots, W_L\}$ remains frozen during gradient descent and will only be updated when the model updates $U(V)^\top$ are manually added. Each pair of $U_l \in \mathbb{R}^{m \times r}$ and $V_l \in \mathbb{R}^{n \times r}$ denotes the low-rank updates of the corresponding frozen parameters $W_l \in \mathbb{R}^{m \times n}$. Throughout the entire training process, the recovered parameters can be expressed as

$$\mathbf{w} = \{W_1, \ldots, W_\rho, W_{\rho+1} + U_{\rho+1}(V_{\rho+1})^\top, \ldots, W_L + U_L(V_L)^\top\}. \tag{13}$$

Let $h(\cdot)$ denote the local objective function for training with low-rank model updates, the goal of FL with FedMUD becomes

$$\min h(\mathbf{x}) \triangleq \frac{1}{N} \sum_{i=1}^{N} h_i(\mathbf{x}), \tag{14}$$

where $h_i(\mathbf{x}) \triangleq \mathbb{E}_{\xi_i \in \mathbb{D}_i}[H_i(\mathbf{x}, \xi_i)]$ is the excepted loss function of client $i$ with low-rank model updates.

Let $\mathbf{x}_i^t$ denote the model parameters of client $i$ at iteration $t$ and $E \ll T$ denote the number of iterations between successive aggregations, we have

$$\mathbf{x}_i^t = \begin{cases} \mathbf{x}_i^{t-1} - \eta \frac{1}{N} \sum_{j=1}^N \nabla H_j(\mathbf{x}_j^{t-1}, \xi_j^t) & \text{if } t \mod E = 0, \\ \mathbf{x}_i^{t-1} - \eta \nabla H_i(\mathbf{x}_i^{t-1}, \xi_i^t) & \text{otherwise}, \end{cases} \tag{15}$$

Next, we define $S \triangleq nE$ to be the number of iterations between consecutive resets of low-rank updates. Specifically, for every $n$ rounds or $S$ iterations, low-rank model updates $U(V)^\top$ will be manually added to frozen parameters. Subsequently, $U$ is reinitialized randomly and $V$ is reinitialized to zero, ensuring that the values of the recovered model parameters $\mathbf{w}$ remain unchanged before and after the reset. In particular, the initialized value of $U$ must remain consistent between different clients, which can be ensured by passing a random seed from the server to clients. This reset process is defined as

$$\mathbf{x}_i^t = \begin{cases} \text{reset\_updates}(\mathbf{x}_i^t) & \text{if } t \mod S = 0, \\ \mathbf{x}_i^t & \text{otherwise}. \end{cases} \tag{16}$$

*Table 6.* Commonly used notations and descriptions.

| Notation | | Description |
|---|---|---|
| $\eta$ | | The learning rate. |
| $i$ | $N$ | The index and total number of clients. |
| $t$ | $T$ | The index and total number of training iterations. |
| $E$ | $S$ | The number of iterations for model aggregation and resetting model updates. |
| $t_e$ | $t_s$ | The iteration indices of last model aggregation and last resetting model updates. |
| $l$ | $L$ | The index and total number of the layers within a neural network. |
| $\rho$ | | The first $\rho$ layers of the network remain the same without applying low-rank model updates. |
| $\mathbf{x}$ | $\mathbf{w}$ | The model parameters and recovered model parameters as defined in Eq.(12) and Eq.(13). |
| $\bar{\mathbf{x}}$ | $\bar{\mathbf{w}}$ | The average results of $\mathbf{x}$ and $\mathbf{w}$ of all clients as defined in Eq.(17) and Eq.(18). |
| $\|\cdot\|_F$ | $\|\cdot\|_2$ | Frobenius norm of a matrix and $\ell_2$ norm of a vector. |
| $h_i$ | $H_i$ | The objective function and its expectation of client $i$, defined in Eq. (14), taking $\mathbf{x}$ as their input. |
| $f_i$ | $F_i$ | The objective function and its expectation of client $i$, defined in Eq. (11), taking $\mathbf{w}$ as their input. |

For the sake of convenience in subsequent analysis, we give the following definitions.

$$\bar{W}_l^t \triangleq \frac{1}{N} \sum_{i=1}^N W_{i,l}^t, \quad \bar{U}_l^t \triangleq \frac{1}{N} \sum_{i=1}^N U_{i,l}^t, \quad \bar{V}_l^t \triangleq \frac{1}{N} \sum_{i=1}^N V_{i,l}^t,$$

$$\bar{\mathbf{x}}^t \triangleq \frac{1}{N} \sum_{i=1}^N \mathbf{x}_i^t = \{\bar{W}_1^t, \ldots, \bar{W}_\rho^t, \bar{W}_{\rho+1}^t, \bar{U}_{\rho+1}^t, \bar{V}_{\rho+1}^t, \ldots, \bar{W}_L^t, \bar{U}_L^t, \bar{V}_L^t\}. \tag{17}$$

Further, we can use $\bar{\mathbf{x}}$ to get the recovered parameters $\bar{\mathbf{w}}$ as follows.

$$\bar{\mathbf{w}}^t = \{\bar{W}_1^t, \ldots, \bar{W}_\rho^t, \bar{W}_{\rho+1}^t + \bar{U}_{\rho+1}^t (\bar{V}_{\rho+1}^t)^\top, \ldots, \bar{W}_L^t + \bar{U}_L^t (\bar{V}_L^t)^\top\} \tag{18}$$

According to the definition of $\bar{\mathbf{x}}$, we have the following property for $\bar{\mathbf{x}}$.

$$\bar{\mathbf{x}}^t = \bar{\mathbf{x}}^{t-1} - \eta \frac{1}{N} \sum_{i=1}^N \nabla H_i(\mathbf{x}_i^{t-1}) \tag{19}$$

Since $\mathbf{w}$ is recovered from $\mathbf{x}$ and $\bar{\mathbf{w}}$ is recovered from $\bar{\mathbf{x}}$, we have $F_i(\mathbf{w}, \xi) = H_i(\mathbf{x}, \xi)$ and $F_i(\bar{\mathbf{w}}, \xi) = H_i(\bar{\mathbf{x}}, \xi)$. Next, we discuss the relationship between the derivatives of $F(\cdot)$ and $H(\cdot)$. We use the subscript $l$ to indicate the trainable parameters and corresponding gradients of layer $l$. For each layer $l \le \rho$, we have $\nabla F_{i,l}^t(\mathbf{w}_i^t, \xi_i^t) = \nabla H_{i,l}^t(\mathbf{x}_i^t, \xi_i^t)$, since $\mathbf{x}_{i,l}^t = \mathbf{w}_{i,l}^t = W_{i,l}^t$. For each layer $l > \rho$, $\nabla H(\cdot)$ and $\nabla F(\cdot)$ satisfy

$$\nabla H_{i,l}(\mathbf{x}_i^t, \xi_i^{t+1}) = \begin{bmatrix} \nabla F_{i,l}(\mathbf{w}_i^t, \xi_i^{t+1}) V_{i,l}^t \\ \nabla F_{i,l}(\mathbf{w}_i^t, \xi_i^{t+1})^\top U_{i,l}^t \end{bmatrix}. \tag{20}$$

Note that we will omit $\xi$ in $F(\cdot)$ and $H(\cdot)$ for ease of writing in the following analysis. Considering that $h_i(\mathbf{x}) = \mathbb{E}[H_i(\mathbf{x})]$ and $\nabla h(\mathbf{x}) = \frac{1}{N} \sum_{i=1}^{N} \nabla h_i(\mathbf{x})$, we have

$$\nabla h_{i,l}(\mathbf{x}_i^t) = \begin{bmatrix} \nabla f_{i,l}(\mathbf{w}_i^t) V_{i,l}^t \\ \nabla f_{i,l}(\mathbf{w}_i^t)^\top U_{i,l}^t \end{bmatrix}, \quad \nabla H_l(\bar{\mathbf{x}}^t) = \begin{bmatrix} \nabla F_l(\bar{\mathbf{w}}^t) \bar{V}_l^t \\ \nabla F_l(\bar{\mathbf{w}}^t)^\top \bar{U}_l^t \end{bmatrix}, \quad \nabla h_l(\bar{\mathbf{x}}^t) = \begin{bmatrix} \nabla f_l(\bar{\mathbf{w}}^t) \bar{V}_l^t \\ \nabla f_l(\bar{\mathbf{w}}^t)^\top \bar{U}_l^t \end{bmatrix}. \tag{21}$$

The Frobenius norm of $\nabla H(\cdot)$ and $\nabla F(\cdot)$ has the following relationship

$$\begin{aligned} \left\| \nabla H_{i,l}(\mathbf{x}_i^{t-1}) \right\|_F^2 &= \left\| \nabla F_{i,l}(\mathbf{w}_i^{t-1}) V_{i,l}^t \right\|_F^2 + \left\| \nabla F_{i,l}(\mathbf{w}_i^{t-1})^\top U_{i,l}^t \right\|_F^2, \\ \left\| \nabla h_{i,l}(\mathbf{x}_i^{t-1}) \right\|_F^2 &= \left\| \nabla f_{i,l}(\mathbf{w}_i^{t-1}) V_{i,l}^t \right\|_F^2 + \left\| \nabla f_{i,l}(\mathbf{w}_i^{t-1})^\top U_{i,l}^t \right\|_F^2. \end{aligned} \tag{22}$$

Finally, the whole gradient of $\mathbf{x}_i^{t-1}$ and $\mathbf{w}_i^{t-1}$ can be represented as

$$\begin{aligned} \nabla h_i(\mathbf{x}_i^{t-1}) &= \{\nabla h_i^1(\mathbf{x}_i^{t-1}), \ldots, \nabla h_i^L(\mathbf{x}_i^{t-1})\}, \quad \nabla H_i(\mathbf{x}_i^{t-1}) = \{\nabla H_i^1(\mathbf{x}_i^{t-1}), \ldots, \nabla H_i^L(\mathbf{x}_i^{t-1})\}, \\ \nabla f_i(\mathbf{w}_i^{t-1}) &= \{\nabla f_i^1(\mathbf{w}_i^{t-1}), \ldots, \nabla f_i^L(\mathbf{w}_i^{t-1})\}, \quad \nabla F_i(\mathbf{w}_i^{t-1}) = \{\nabla F_i^1(\mathbf{w}_i^{t-1}), \ldots, \nabla F_i^L(\mathbf{w}_i^{t-1})\}, \end{aligned} \tag{23}$$

where $\nabla h_i(\mathbf{x}_i^{t-1})$, $\nabla H_i(\mathbf{x}_i^{t-1})$, $\nabla f_i(\mathbf{w}_i^{t-1})$ and $\nabla F_i(\mathbf{w}_i^{t-1})$ are flattened vectors, and we have

$$\begin{aligned} \left\| \nabla h_i(\mathbf{x}_i^{t-1}) \right\|_2^2 &= \sum_{l=1}^{L} \left\| \nabla h_{i,l}(\mathbf{x}_i^{t-1}) \right\|_F^2, \quad \left\| \nabla H_i(\mathbf{x}_i^{t-1}) \right\|_2^2 = \sum_{l=1}^{L} \left\| \nabla H_{i,l}(\mathbf{x}_i^{t-1}) \right\|_F^2, \\ \left\| \nabla f_i(\mathbf{w}_i^{t-1}) \right\|_2^2 &= \sum_{l=1}^{L} \left\| \nabla f_{i,l}(\mathbf{w}_i^{t-1}) \right\|_F^2, \quad \left\| \nabla F_i(\mathbf{w}_i^{t-1}) \right\|_2^2 = \sum_{l=1}^{L} \left\| \nabla F_{i,l}(\mathbf{w}_i^{t-1}) \right\|_F^2. \end{aligned} \tag{24}$$

In the following, we demonstrate that the sequence of recovered model parameters $\{\mathbf{w}^1, \mathbf{w}^2, \ldots, \mathbf{w}^t, \ldots, \mathbf{w}^T\}$, obtained by FedMUD, converges to a local stationary point in the optimization space of $\mathbf{w}$ under non-convex and smooth assumptions. The commonly used notations are summarized in Table 6, where $t_e$ and $t_s$ are used to represent the iteration indices of last aggregation and last resetting model updates. Thus, we have $0 \le t - t_e < E$ and $0 \le t - t_s < S$.

### C.2. Assumptions

**Assumption 1.** *The loss functions $f_i(\cdot)$ and $h_i(\cdot)$ are differentiable and $L$-smooth, with a constant $L_s$.*

$$\begin{aligned} f_i(\mathbf{w}_1) &\le f_i(\mathbf{w}_2) + \langle \mathbf{w}_1 - \mathbf{w}_2, \nabla f_i(\mathbf{w}_2) \rangle + \frac{L_s}{2} \|\mathbf{w}_1 - \mathbf{w}_2\|_2^2, \quad \forall i, \forall \mathbf{w}_1, \forall \mathbf{w}_2, \\ h_i(\mathbf{x}_1) &\le h_i(\mathbf{x}_2) + \langle \mathbf{x}_1 - \mathbf{x}_2, \nabla h_i(\mathbf{x}_2) \rangle + \frac{L_s}{2} \|\mathbf{x}_1 - \mathbf{x}_2\|_2^2, \quad \forall i, \forall \mathbf{x}_1, \forall \mathbf{x}_2. \end{aligned} \tag{25}$$

**Assumption 2.** *The stochastic gradient $\nabla F_i(\mathbf{w}_i^t, \xi_i^t)$ and $\nabla H_i(\mathbf{x}_i^t, \xi_i^t)$ are unbiased, with bounded variance and norm.*

$$\mathbb{E}_{\xi_i \in \mathbb{D}_i} \nabla F_i(\mathbf{w}_i^t, \xi_i^t) = \nabla f_i(\mathbf{w}_i^t), \quad \mathbb{E}_{\xi_i \in \mathbb{D}_i} \nabla H_i(\mathbf{x}_i^t, \xi_i^t) = \nabla h_i(\mathbf{x}_i^t), \quad \forall i, \forall t$$

$$\mathbb{E}_{\xi_i \in \mathbb{D}_i} \left\| \nabla F_i(\mathbf{w}_i^t, \xi_i^t) - \nabla f_i(\mathbf{w}_i^t) \right\|_2^2 \le \sigma^2, \quad \mathbb{E}_{\xi_i \in \mathbb{D}_i} \left\| \nabla F_{i,l}(\mathbf{w}_i^t, \xi_i^t) \right\|_F^2 \le G^2, \quad \forall i, \forall t, \forall l \tag{26}$$

**Assumption 3.** *The Frobenius norm of matrices $U$ and $V$ is bounded during both initialization and training.*

$$\left\| U_{i,l}^t \right\|_F^2 \le \kappa_u^2, \quad \left\| V_{i,l}^t \right\|_F^2 \le \kappa_v^2, \quad and \quad \left\| U_l^{t_s} \right\|_F^2 \le \epsilon_u^2 \ll \kappa_u^2, \quad \left\| U_l^{t_s} \right\|_F^2 \le \epsilon_v^2 \ll \kappa_v^2, \quad \forall i, \forall t, \forall l > \rho \tag{27}$$

**Assumption 4.** *At least one of the matrices in $\{\bar{U}_l, \bar{V}_l\}$ has a smallest singular value greater than zero.*

$$[\delta_{min}(\bar{U}_l^t)]^2 + [\delta_{min}(\bar{V}_l^t)]^2 \ge \psi_{uv}^2 > 0, \quad \forall t, \forall l > \rho, \tag{28}$$

where $\bar{U}_l^t = \frac{1}{N} \sum_{i=1}^{N} U_{i,l}^t$, $\bar{V}_l^t = \frac{1}{N} \sum_{i=1}^{N} V_{i,l}^t$ and $\delta_{min}(\cdot)$ return the smallest singular value of a matrix.

Note that Assumption 3 is reasonable and commonly adopted in convergence analysis, as discussed in Section 4. In FedMUD, the low-rank matrices $U$ and $V$ undergo local gradient descent updates for a finite number of steps ($\tau$). Following this, they are merged into the base model parameters and subsequently reinitialized. This periodic reinitialization mechanism prevents the unbounded growth of $U$ and $V$. To illustrate, for any matrix $U$, its value after $t$ steps ($t \le \tau$) is $U_t = U_0 + \eta \sum_{i=1}^{t} g_i$. Given $\|g_i\|_F \le G$, its squared norm is bounded by $\|U_t\|_F^2 \le (\tau + 1) \left[ \|U_0\|_F^2 + \eta^2 \tau^2 G^2 \right]$. Therefore, the combination of finite local training and periodic reinitialization ensures that $U$ and $V$ remain bounded, validating our assumption.

### C.3. Lemmas

Before presenting the lemmas, we first introduce an inequality as follows

$$\left\|\sum_{i=1}^{n}\mathbf{z}_i\right\|_2^2 \leq n\sum_{i=1}^{n}\|\mathbf{z}_i\|_2^2, \tag{29}$$

which holds for any vector $\mathbf{z}_i$ and any positive integer $n$. This inequality also applies to the Frobenius norm of matrices and can be seen as a special case of the Minkowski inequality. It will be used frequently in the subsequent analysis. For convenience, we will refer to it as the Minkowski inequality in the following discussion, without further elaboration.

**Lemma 1.** *Under Assumption 4, at each iteration $t$ and for each layer $l > \rho$, it follows that*

$$\left\|\nabla h_l(\mathbf{x}^t)\right\|_F^2 \geq \psi_{uv}^2 \left\|\nabla f_l(\mathbf{w}^t)\right\|_F^2. \tag{30}$$

**Proof of Lemma 1.** According to Eq.(21), for each layer $l > \rho$, we have

$$\begin{aligned}
\left\|\nabla h_l(\mathbf{x}^t)\right\|_F^2 &= \left\|\nabla f_l(\mathbf{w}^t)V_l^t\right\|_F^2 + \left\|(\nabla f_l(\mathbf{w}^t))^\top U_l^t\right\|_F^2 \\
&= \left\|(V_l^t)^\top \nabla f_l(\mathbf{w}^t)^\top\right\|_F^2 + \left\|(U_l^t)^\top \nabla f_l(\mathbf{w}^t)\right\|_F^2 \\
&\overset{(a)}{\geq} (\delta_{\min}^2(V_l^t) + \delta_{\min}^2(U_l^t)) \left\|\nabla f_l(\mathbf{w}^t)\right\|_F^2, \\
&\overset{(b)}{\geq} \psi_{uv}^2 \left\|\nabla f_l(\mathbf{w}^t)\right\|_F^2
\end{aligned} \tag{31}$$

where (a) follows from the inequality $\delta_{\min}(U)\|V\|_F \leq \|UV\|_F$ for any matrix $U \in \mathbb{R}^{m\times r}$ and matrix $V \in \mathbb{R}^{r\times n}$. The proof of this inequality can be found in Lemma B.3 in (Zou et al., 2020). (b) follows from Eq.(28) in Assumption 4.

**Lemma 2.** *Under Assumptions 2 and 3, at each iteration $t$ and for every client $i$ and each layer $l > \rho$, it follows that*

$$\mathbb{E}\left[\left\|U_{i,l}^t\right\|_F^2\right] \leq \Gamma_u^2 \triangleq \min\{2\epsilon_u^2 + 2\eta^2 S^2 G^2 \kappa_v^2, \kappa_u^2\}, \quad \mathbb{E}\left[\left\|V_{i,l}^t\right\|_F^2\right] \leq \Gamma_v^2 \triangleq \min\{2\epsilon_v^2 + 2\eta^2 S^2 G^2 \kappa_u^2, \kappa_v^2\}. \tag{32}$$

**Proof of Lemma 2.** Based on the relationship between $t$ and $t_s$, we consider two cases. First, when $t = t_s$, the matrix $U$ is reinitialized. In this case, by Assumption 3, we have $\mathbb{E}\left[\left\|U_{i,l}^t\right\|_F^2\right] = \mathbb{E}\left[\left\|U_{i,l}^{t_s}\right\|_F^2\right] \leq \epsilon_u^2 \leq \Gamma_u^2$. Next, for the case where $t > t_s$, the update of $U$ after initialization can be divided into aggregated gradients and unaggregated gradients as follows

$$U_{i,l}^t = U_{i,l}^{t_s} - \eta\sum_{\tau=t_s+1}^{t_e}\frac{1}{N}\sum_{j=1}^{N}\nabla F_{j,l}(\mathbf{w}_j^{\tau-1})V_{j,l}^{\tau-1} - \eta\sum_{\tau=t_e+1}^{t}\nabla F_{i,l}(\mathbf{w}_i^{\tau-1})V_{i,l}^{\tau-1} \tag{33}$$

Thus, we have

$$\begin{aligned}
&\mathbb{E}\left[\left\|U_{i,l}^t\right\|_F^2\right] \\
&\overset{(a)}{\leq} 2\mathbb{E}\left[\left\|U_{i,l}^{t_s}\right\|_F^2 + \eta^2\left\|\sum_{\tau=t_s+1}^{t_e}\frac{1}{N}\sum_{j=1}^{N}\nabla F_{j,l}(\mathbf{w}_j^{\tau-1})V_{j,l}^{\tau-1} + \sum_{\tau=t_e+1}^{t}\nabla F_{i,l}(\mathbf{w}_i^{\tau-1})V_{i,l}^{\tau-1}\right\|_F^2\right] \\
&\overset{(b)}{\leq} 2\mathbb{E}\left\|U_{i,l}^{t_s}\right\|_F^2 + 2\eta^2(t-t_s)\mathbb{E}\left[\sum_{\tau=t_s+1}^{t_e}\left\|\frac{1}{N}\sum_{j=1}^{N}\nabla F_{j,l}(\mathbf{w}_j^{\tau-1})V_{j,l}^{\tau-1}\right\|_F^2 + \sum_{\tau=t_e+1}^{t}\left\|\nabla F_{i,l}(\mathbf{w}_i^{\tau-1})V_{i,l}^{\tau-1}\right\|_F^2\right] \\
&\overset{(c)}{\leq} 2\mathbb{E}\left\|U_{i,l}^{t_s}\right\|_F^2 + 2\eta^2(t-t_s)\mathbb{E}\left[\sum_{\tau=t_s+1}^{t_e}\frac{1}{N}\sum_{j=1}^{N}\left\|\nabla F_{j,l}(\mathbf{w}_j^{\tau-1})V_{j,l}^{\tau-1}\right\|_F^2 + \sum_{\tau=t_e+1}^{t}\left\|\nabla F_{i,l}(\mathbf{w}_i^{\tau-1})V_{i,l}^{\tau-1}\right\|_F^2\right] \\
&\overset{(d)}{\leq} 2\mathbb{E}\left\|U_{i,l}^{t_s}\right\|_F^2 + 2\eta^2(t-t_s)^2 G^2\kappa_v^2 \\
&\overset{(e)}{\leq} 2\epsilon_u^2 + 2\eta^2 S^2 G^2 \kappa_v^2,
\end{aligned} \tag{34}$$

where (a), (b) and (c) result from applying the Minkowski inequality in Eq.(29) with $n = 2$, $t - t_s$ and $N$, respectively. (d) follows from the basic inequality $\|PQ\|_F^2 \leq \|P\|_F^2 \cdot \|Q\|_F^2$, Eq.(26) in Assumption 2 and Eq.(27) in Assumption 3. (e) follows from the inequality $(t - t_s) < S$ and Eq.(27) in Assumption 3.

Furthermore, by Eq.(27), we have $\mathbb{E}\left[\left\|U_{i,l}^t\right\|_F^2\right] \leq \min\{2\epsilon_u^2 + 2\eta^2 S^2 G^2 \kappa_v^2, \kappa_u^2\}$. Considering the symmetry of the matrices $U$ and $V$, we can repeat the above process and have $\mathbb{E}\left[\left\|V_{i,l}^t\right\|_F^2\right] \leq \min\{2\epsilon_v^2 + 2\eta^2 S^2 G^2 \kappa_u^2, \kappa_v^2\}$.

**Lemma 3.** *Under Assumptions 1 and 2, at each iteration $t$ and for every client $i$ and each layer $l > \rho$, it follows that*

$$\mathbb{E}\left[\left\|\bar{\mathbf{x}}_l^t - \mathbf{x}_{i,l}^t\right\|_F^2\right] \leq 4\eta^2 E^2 G^2 (\Gamma_u^2 + \Gamma_v^2) \tag{35}$$

**Proof of Lemma 3.** Similar to Lemma 2, we analyze two cases based on the relationship between $t$ and $t_e$. First, for the case $t = t_e$, all model parameters are aggregated. Thus, we have $\mathbf{x}_{i,l}^t = \bar{\mathbf{x}}_l^t$ and $\mathbb{E}\left[\left\|\bar{\mathbf{x}}_l^t - \mathbf{x}_{i,l}^t\right\|_F^2\right] = 0 \leq 4\eta^2 E^2 G^2 (\Gamma_u^2 + \Gamma_v^2)$. Next, for the case $t > t_s$, both $\mathbf{x}_i^t$ and $\bar{\mathbf{x}}^t$ can be represented using $\bar{\mathbf{x}}^{t_e}$ as follows

$$\mathbf{x}_{i,l}^t = \bar{\mathbf{x}}_l^{t_e} - \eta \sum_{\tau=t_e+1}^t \nabla H_{i,l}(\mathbf{x}_i^{\tau-1}), \quad \bar{\mathbf{x}}_l^t = \bar{\mathbf{x}}_l^{t_e} - \eta \sum_{\tau=t_e+1}^t \frac{1}{N} \sum_{i=1}^N \nabla H_{i,l}(\mathbf{x}_i^{\tau-1}). \tag{36}$$

Thus, we have

$$
\begin{aligned}
\mathbb{E}\left[\left\|\bar{\mathbf{x}}_l^t - \mathbf{x}_{i,l}^t\right\|_F^2\right] &= \mathbb{E}\left[\left\|\eta \sum_{\tau=t_e+1}^t \frac{1}{N} \sum_{i=1}^N \nabla H_{i,l}(\mathbf{x}_i^{\tau-1}) - \eta \sum_{\tau=t_e+1}^t \nabla H_{i,l}(\mathbf{x}_i^{\tau-1})\right\|_F^2\right] \\
&\stackrel{(a)}{\leq} 2\eta^2 (t - t_e) \mathbb{E}\left[\sum_{\tau=t_e+1}^t \left\|\frac{1}{N} \sum_{i=1}^N \nabla H_{i,l}(\mathbf{x}_i^{\tau-1})\right\|_F^2 + \sum_{\tau=t_e+1}^t \left\|\nabla H_{i,l}(\mathbf{x}_i^{\tau-1})\right\|_F^2\right] \\
&\stackrel{(b)}{\leq} 2\eta^2 (t - t_e) \mathbb{E}\left[\frac{1}{N} \sum_{i=1}^N \sum_{\tau=t_e+1}^t \left\|\nabla H_{i,l}(\mathbf{x}_i^{\tau-1})\right\|_F^2 + \sum_{\tau=t_e+1}^t \left\|\nabla H_{i,l}(\mathbf{x}_i^{\tau-1})\right\|_F^2\right] \\
&\stackrel{(c)}{\leq} 4\eta^2 (t - t_e)^2 \mathbb{E}\left[\left\|\nabla F_{i,l}(\mathbf{w}_i^{\tau-1}) V_{i,l}^{\tau-1}\right\|_F^2 + \left\|\nabla F_{i,l}(\mathbf{w}_i^{\tau-1})^\top U_{i,l}^{\tau-1}\right\|_F^2\right], \\
&\stackrel{(d)}{\leq} 4\eta^2 E^2 G^2 (\Gamma_u^2 + \Gamma_v^2),
\end{aligned}
\tag{37}
$$

where (a) and (b) result from applying the Minkowski inequality in Eq.(29) with $n = 2(t - t_e)$ and $N$, respectively. (c) follows from Eq.(22). (d) follows from inequality $\|PQ\|_F^2 \leq \|P\|_F^2 \cdot \|Q\|_F^2$, Assumption 2 and in Lemma 2.

**Lemma 4.** *Under Assumptions 1, 2 and 3, at each iteration $t$ and for every client $i$, it follows that*

$$\mathbb{E}\left[\left\|\bar{\mathbf{w}}^t - \mathbf{w}_i^t\right\|_2^2\right] \leq \Lambda \triangleq 4\rho\eta^2 E^2 G^2 + 12(L - \rho)\eta^2 E^2 G^2 (\Gamma_u^4 + \Gamma_v^4 + \eta^2 E^2 G^2 \Gamma_u^2 \Gamma_v^2). \tag{38}$$

**Proof of Lemma 4.** Similar to Lemma 3, we analyze two cases based on the relationship between $t$ and $t_e$. First, for the case $t = t_e$, $\mathbf{w}_i^t = \bar{\mathbf{w}}^t$, and thus $\mathbb{E}\left[\left\|\bar{\mathbf{w}}^t - \mathbf{w}_i^t\right\|_2^2\right] = 0 \leq \Lambda$. Next, for the case $t > t_e$, we can divide the model parameters into full and low-rank components and compute $\mathbb{E}\left[\left\|\bar{\mathbf{w}}^t - \mathbf{w}_i^t\right\|_2^2\right]$ as follows

$$\mathbb{E}\left[\left\|\bar{\mathbf{w}}^t - \mathbf{w}_i^t\right\|_2^2\right] = \mathbb{E}\left[\sum_{l=1}^\rho \left\|\bar{W}_l^t - W_{i,l}^t\right\|_F^2 + \sum_{l=\rho+1}^L \left\|\bar{U}_l^t (\bar{V}_l^t)^\top - U_{i,l}^t (V_{i,l}^t)^\top\right\|_F^2\right]. \tag{39}$$

For the first consider the term $\mathbb{E}\sum_{l=1}^\rho \left\|\bar{W}_l^t - W_{i,l}^t\right\|_F^2$, we have

$$W_{i,l}^t = \bar{W}_l^{t_e} - \eta \sum_{\tau=t_e+1}^t \nabla F_{i,l}(\mathbf{w}_i^{\tau-1}), \quad \bar{W}_l^t = \bar{W}_l^{t_e} - \eta \sum_{\tau=t_e+1}^t \frac{1}{N} \sum_{j=1}^N \nabla F_{j,l}(\mathbf{w}_j^{\tau-1}). \tag{40}$$

Thus, we have

$$
\begin{aligned}
\mathbb{E}\left[\sum_{l=1}^{\rho}\left\|\bar{W}_l^t - W_{i,l}^t\right\|_F^2\right] &= \mathbb{E}\left[\sum_{l=1}^{\rho}\left\|\eta\sum_{\tau=t_e+1}^{t}\frac{1}{N}\sum_{j=1}^{N}\nabla F_{j,l}(\mathbf{w}_j^{\tau-1}) - \eta\sum_{\tau=t_e+1}^{t}\nabla F_{i,l}(\mathbf{w}_i^{\tau-1})\right\|_F^2\right] \\
&\overset{(a)}{\leq}\sum_{l=1}^{\rho}2\eta^2(t-t_e)\sum_{\tau=t_e+1}^{t}\mathbb{E}\left[\left\|\frac{1}{N}\sum_{j=1}^{N}\nabla F_{j,l}(\mathbf{w}_j^{\tau-1})\right\|_F^2 + \left\|\nabla F_{i,l}(\mathbf{w}_i^{\tau-1})\right\|_F^2\right] \\
&\overset{(b)}{\leq}\sum_{l=1}^{\rho}2\eta^2(t-t_e)\sum_{\tau=t_e+1}^{t}\mathbb{E}\left[\frac{1}{N}\sum_{j=1}^{N}\left\|\nabla F_{j,l}(\mathbf{w}_j^{\tau-1})\right\|_F^2 + \left\|\nabla F_{i,l}(\mathbf{w}_i^{\tau-1})\right\|_F^2\right] \\
&\overset{(c)}{\leq}\sum_{l=1}^{\rho}4\eta^2(t-t_e)^2 G^2 \overset{(d)}{\leq} 4\rho\eta^2 E^2 G^2,
\end{aligned}
\tag{41}
$$

where (a) and (b) follow from the Minkowski inequality. (c) follows from Assumption 2. (d) follows from $t - t_e < E$.

For the second term $\mathbb{E}\left[\sum_{l=\rho+1}^{L}\left\|\bar{U}_l^t(\bar{V}_l^t)^\top - U_{i,l}^t(V_{i,l}^t)^\top\right\|_F^2\right]$, we have

$$
\begin{aligned}
U_{i,l}^t &= \bar{U}_l^{t_e} - \eta\sum_{\tau=t_e+1}^{t}\nabla F_{i,l}(\mathbf{w}_i^{\tau-1})V_{i,l}^{\tau-1}, \quad \bar{U}_l^t = \bar{U}_l^{t_e} - \eta\sum_{\tau=t_e+1}^{t}\frac{1}{N}\sum_{j=1}^{N}\nabla F_{j,l}(\mathbf{w}_j^{\tau-1})V_{j,l}^{\tau-1} \\
V_{i,l}^t &= \bar{V}_l^{t_e} - \eta\sum_{\tau=t_e+1}^{t}\nabla F_{i,l}(\mathbf{w}_i^{\tau-1})^\top U_{i,l}^{\tau-1}, \quad \bar{V}_l^t = \bar{V}_l^{t_e} - \eta\sum_{\tau=t_e+1}^{t}\frac{1}{N}\sum_{j=1}^{N}\nabla F_{j,l}(\mathbf{w}_j^{\tau-1})^\top U_{j,l}^{\tau-1}
\end{aligned}
\tag{42}
$$

Thus, we have

$$
\begin{aligned}
\bar{U}_l^t(\bar{V}_l^t)^\top - U_{i,l}^t(V_{i,l}^t)^\top &= \underbrace{-\eta\left[\sum_{\tau=t_e+1}^{t}\frac{1}{N}\sum_{j=1}^{N}\nabla F_{j,l}(\mathbf{w}_j^{\tau-1})V_{j,l}^{\tau-1} - \sum_{\tau=t_e+1}^{t}\nabla F_{i,l}(\mathbf{w}_i^{\tau-1})V_{i,l}^{\tau-1}\right](\bar{V}_l^{t_e})^\top}_{A_l} \\
&\quad \underbrace{-\eta\bar{U}_l^{t_e}\left[\sum_{\tau=t_e+1}^{t}\frac{1}{N}\sum_{j=1}^{N}(U_{j,l}^{\tau-1})^\top\nabla F_{j,l}(\mathbf{w}_j^{\tau-1}) - \sum_{\tau=t_e+1}^{t}(U_{i,l}^{\tau-1})^\top\nabla F_{i,l}(\mathbf{w}_i^{\tau-1})\right]}_{B_l} \\
&\quad \underbrace{+\eta^2\left[\sum_{\tau=t_e+1}^{t}\frac{1}{N}\sum_{j=1}^{N}\nabla F_{j,l}(\mathbf{w}_j^{\tau-1})V_{j,l}^{\tau-1}\right]\left[\sum_{\tau=t_e+1}^{t}\frac{1}{N}\sum_{j=1}^{N}(U_{j,l}^{\tau-1})^\top\nabla F_{j,l}(\mathbf{w}_j^{\tau-1})\right]}_{C_l} \\
&\quad \underbrace{-\eta^2\left[\sum_{\tau=t_e+1}^{t}\nabla F_{i,l}(\mathbf{w}_i^{\tau-1})V_{i,l}^{\tau-1}\right]\left[\sum_{\tau=t_e+1}^{t}(U_{i,l}^{\tau-1})^\top\nabla F_{i,l}(\mathbf{w}_i^{\tau-1})\right]}_{D_l}
\end{aligned}
\tag{43}
$$

Next, we try to find the upper bound of $\mathbb{E}\left[\|A_l\|_F^2\right]$, $\mathbb{E}\left[\|B_l\|_F^2\right]$, $\mathbb{E}\left[\|C_l\|_F^2\right]$, and $\mathbb{E}\left[\|D_l\|_F^2\right]$, respectively.

$$\mathbb{E}\left[\|A_l\|_F^2\right] = \eta^2 \mathbb{E}\left\|\left[\sum_{\tau=t_e+1}^{t}\frac{1}{N}\sum_{j=1}^{N}\nabla F_{j,l}(\mathbf{w}_j^{\tau-1})V_{j,l}^{\tau-1} - \sum_{\tau=t_e+1}^{t}\nabla F_{i,l}(\mathbf{w}_i^{\tau-1})V_{i,l}^{\tau-1}\right](\bar{V}_l^{t_e})^\top\right\|_F^2$$

$$\overset{(a)}{\leq}\eta^2\Gamma_v^2\mathbb{E}\left\|\sum_{\tau=t_e+1}^{t}\frac{1}{N}\sum_{j=1}^{N}\nabla F_{j,l}(\mathbf{w}_j^{\tau-1})V_{j,l}^{\tau-1} - \sum_{\tau=t_e+1}^{t}\nabla F_{i,l}(\mathbf{w}_i^{\tau-1})V_{i,l}^{\tau-1}\right\|_F^2$$

$$\overset{(b)}{\leq}2\eta^2\Gamma_v^2(t-t_e)\sum_{\tau=t_e+1}^{t}\mathbb{E}\left[\left\|\frac{1}{N}\sum_{j=1}^{N}\nabla F_{j,l}(\mathbf{w}_j^{\tau-1})V_{j,l}^{\tau-1}\right\|_F^2 + \left\|\nabla F_{i,l}(\mathbf{w}_i^{\tau-1})V_{i,l}^{\tau-1}\right\|_F^2\right] \tag{44}$$

$$\overset{(c)}{\leq}2\eta^2\Gamma_v^2(t-t_e)\sum_{\tau=t_e+1}^{t}\mathbb{E}\left[\frac{1}{N}\sum_{j=1}^{N}\left\|\nabla F_{j,l}(\mathbf{w}_j^{\tau-1})V_{j,l}^{\tau-1}\right\|_F^2 + \left\|\nabla F_{i,l}(\mathbf{w}_i^{\tau-1})V_{i,l}^{\tau-1}\right\|_F^2\right]$$

$$\overset{(d)}{\leq}4\eta^2(t-t_e)^2G^2\Gamma_v^4 \leq 4\eta^2E^2G^2\Gamma_v^4,$$

where (a) and (d) follow from the inequality $\|PQ\|_F^2 \leq \|P\|_F^2 \cdot \|Q\|_F^2$. (b) and (c) result from applying the Minkowski inequality in Eq.(29) with $n = 2(t-t_e)$ and $N$, respectively. Note that $A_l$ and $B_l$ are symmetric, and similarly, we can deduce $\mathbb{E}\left[\|B_l\|_F^2\right] \leq 4\eta^2E^2G^2\Gamma_u^4$. Next, for $C_l$, we have

$$\mathbb{E}\left[\|C_l\|_F^2\right] = \eta^4\mathbb{E}\left\|\left[\sum_{\tau=t_e+1}^{t}\frac{1}{N}\sum_{j=1}^{N}\nabla F_{j,l}(\mathbf{w}_j^{\tau-1})V_{j,l}^{\tau-1}\right]\left[\sum_{\tau=t_e+1}^{t}\frac{1}{N}\sum_{j=1}^{N}(U_{j,l}^{\tau-1})^\top\nabla F_{j,l}(\mathbf{w}_j^{\tau-1})\right]\right\|_F^2$$

$$\overset{(a)}{\leq}\eta^4\mathbb{E}\left[\left\|\sum_{\tau=t_e+1}^{t}\frac{1}{N}\sum_{j=1}^{N}\nabla F_{j,l}(\mathbf{w}_j^{\tau-1})V_{j,l}^{\tau-1}\right\|_F^2 \cdot \left\|\sum_{\tau=t_e+1}^{t}\frac{1}{N}\sum_{j=1}^{N}(U_{j,l}^{\tau-1})^\top\nabla F_{j,l}(\mathbf{w}_j^{\tau-1})\right\|_F^2\right]$$

$$\overset{(b)}{\leq}\frac{\eta^4(t-t_e)^2}{N^2}\mathbb{E}\left[\sum_{\tau=t_e+1}^{t}\sum_{j=1}^{N}\left\|\nabla F_{j,l}(\mathbf{w}_j^{\tau-1})V_{j,l}^{\tau-1}\right\|_F^2 \cdot \sum_{\tau=t_e+1}^{t}\sum_{j=1}^{N}\left\|(U_{j,l}^{\tau-1})^\top\nabla F_{j,l}(\mathbf{w}_j^{\tau-1})\right\|_F^2\right] \tag{45}$$

$$\overset{(c)}{\leq}\frac{\eta^4(t-t_e)^2}{N^2}\mathbb{E}\left[\sum_{\tau=t_e+1}^{t}\sum_{j=1}^{N}\left[\left\|\nabla F_{j,l}(\mathbf{w}_j^{\tau-1})\right\|_F^2 \cdot \left\|V_{j,l}^{\tau-1}\right\|_F^2\right] \cdot \sum_{\tau=t_e+1}^{t}\sum_{j=1}^{N}\left[\left\|(U_{j,l}^{\tau-1})^\top\right\|_F^2 \cdot \left\|\nabla F_{j,l}(\mathbf{w}_j^{\tau-1})\right\|_F^2\right]\right]$$

$$\overset{(d)}{\leq}\frac{\eta^4(t-t_e)^2}{N^2}\mathbb{E}\left[\sum_{\tau=t_e+1}^{t}\sum_{j=1}^{N}\left[G^2\Gamma_v^2\right] \cdot \sum_{\tau=t_e+1}^{t}\sum_{j=1}^{N}\left[G^2\Gamma_u^2\right]\right] = \eta^4(t-t_e)^4G^4\Gamma_u^2\Gamma_v^2 \leq \eta^4E^4G^4\Gamma_u^2\Gamma_v^2,$$

where (a) and (c) follow from the inequality $\|PQ\|_F^2 \leq \|P\|_F^2 \cdot \|Q\|_F^2$. (b) results from applying the Minkowski inequality with $n = N(t-t_e)$. (d) follows from Assumption 2 and in Lemma 2. Similarly to $\mathbb{E}\left[\|C_l\|_F^2\right]$, we can obtain the same bound for $D_l$, which is $\mathbb{E}\left[\|D_l\|_F^2\right] \leq \eta^4E^4G^4\Gamma_u^2\Gamma_v^2$. Combining the result of $A_l, B_l, C_l$ and $D_l$, we have

$$\mathbb{E}\left[\sum_{l=\rho+1}^{L}\left\|\bar{U}_l^t(\bar{V}_l^t)^\top - U_{i,l}^t(V_{i,l}^t)^\top\right\|_F^2\right] = \mathbb{E}\left[\sum_{l=\rho+1}^{L}\|A_l + B_l + C_l + D_l\|_F^2\right]$$

$$\leq 3\sum_{l=\rho+1}^{L}\mathbb{E}\left[\|A_l\|_F^2 + \|B_l\|_F^2 + \|C_l + D_l\|_F^2\right] \leq 3\sum_{l=\rho+1}^{L}\mathbb{E}\left[\|A_l\|_F^2 + \|B_l\|_F^2 + 2\|C_l\|_F^2 + 2\|D_l\|_F^2\right] \tag{46}$$

$$= 3\sum_{l=\rho+1}^{L}(4\eta^2E^2G^2\Gamma_u^4 + 4\eta^2E^2G^2\Gamma_v^4 + 4\eta^4E^4G^4\Gamma_u^2\Gamma_v^2) = 12(L-\rho)\eta^2E^2G^2(\Gamma_u^4 + \Gamma_v^4 + \eta^2E^2G^2\Gamma_u^2\Gamma_v^2)$$

Finally, adding Eq.(41) and Eq.(46) to Eq.(40) yields the result of Lemma 4.

**Lemma 5.** *Under Assumptions 1, 2 and 3, at each iteration t, it follows that*

$$\mathbb{E}\left[\left\|\bar{\mathbf{w}}^t - \bar{\mathbf{w}}^{t-1}\right\|_2^2\right] \leq \sum_{l=1}^{\rho} \eta^2 \mathbb{E}\left[\left\|\frac{1}{N}\sum_{i=1}^{N}\nabla f_{i,l}(\mathbf{w}_i^{t-1})\right\|_F^2\right] + \frac{\eta^2\sigma^2}{N} + 3(L-\rho)\eta^2 G^2(\Gamma_u^4 + \Gamma_v^4 + \eta^2 G^2 \Gamma_u^2 \Gamma_v^2) \quad (47)$$

**Proof of Lemma 5.** Similar to Lemma 4, we divide the model parameters into full and low-rank components as follows

$$\mathbb{E}\left[\left\|\bar{\mathbf{w}}^t - \bar{\mathbf{w}}^{t-1}\right\|_2^2\right] = \mathbb{E}\left[\sum_{l=1}^{\rho}\left\|\bar{W}_l^t - \bar{W}_l^{t-1}\right\|_F^2\right] + \mathbb{E}\left[\sum_{l=\rho+1}^{L}\left\|\bar{U}_l^t(\bar{V}_l^t)^\top - \bar{U}_l^{t-1}(\bar{V}_l^{t-1})^\top\right\|_F^2\right] \quad (48)$$

For the first term, we have

$$\mathbb{E}\left[\sum_{l=1}^{\rho}\left\|\bar{W}_l^t - \bar{W}_l^{t-1}\right\|_F^2\right] = \sum_{l=1}^{\rho}\eta^2\mathbb{E}\left[\left\|\frac{1}{N}\sum_{i=1}^{N}\nabla F_{i,l}(\mathbf{w}_i^{t-1})\right\|_F^2\right]$$

$$\overset{(a)}{=} \sum_{l=1}^{\rho}\eta^2\mathbb{E}\left[\left\|\frac{1}{N}\sum_{i=1}^{N}\left[\nabla F_{i,l}(\mathbf{w}_i^{t-1}) - \nabla f_{i,l}(\mathbf{w}_i^{t-1})\right]\right\|_F^2\right] + \sum_{l=1}^{\rho}\eta^2\mathbb{E}\left[\left\|\frac{1}{N}\sum_{i=1}^{N}\nabla f_{i,l}(\mathbf{w}_i^{t-1})\right\|_F^2\right]$$

$$= \sum_{l=1}^{\rho}\frac{\eta^2}{N^2}\sum_{i=1}^{N}\mathbb{E}\left[\left\|\nabla F_{i,l}(\mathbf{w}_i^{t-1}) - \nabla f_{i,l}(\mathbf{w}_i^{t-1})\right\|_F^2\right] + \sum_{l=1}^{\rho}\eta^2\mathbb{E}\left[\left\|\frac{1}{N}\sum_{i=1}^{N}\nabla f_{i,l}(\mathbf{w}_i^{t-1})\right\|_F^2\right] \quad (49)$$

$$\overset{(b)}{\leq} \frac{\eta^2\sigma^2}{N} + \sum_{l=1}^{\rho}\eta^2\mathbb{E}\left[\left\|\frac{1}{N}\sum_{i=1}^{N}\nabla f_{i,l}(\mathbf{w}_i^{t-1})\right\|_F^2\right],$$

where both (a) and (c) follow from Assumption 2. (a) and (c) leverage the unbiased and bounded variance properties of the gradient, respectively.

For the second term, we have

$$\bar{U}_l^t(\bar{V}_l^t)^\top - \bar{U}_l^{t-1}(\bar{V}_l^{t-1})^\top = \underbrace{-\eta\frac{1}{N}\sum_{i=1}^{N}\nabla F_{i,l}(\mathbf{w}_i^{t-1})V_{i,l}^{t-1}(\bar{V}_l^{t-1})^\top}_{A_l} \underbrace{-\eta\frac{1}{N}\sum_{i=1}^{N}\bar{U}_l^{t-1}(U_{i,l}^{t-1})^\top\nabla F_{i,l}(\mathbf{w}_i^{t-1})}_{B_l}$$

$$+ \underbrace{\eta^2\left[\frac{1}{N}\sum_{i=1}^{N}\nabla F_{i,l}(\mathbf{w}_i^{t-1})V_{i,l}^{t-1}\right]\cdot\left[\frac{1}{N}\sum_{i=1}^{N}(U_{i,l}^{t-1})^\top\nabla F_{i,l}(\mathbf{w}_i^{t-1})\right]}_{C_l} \quad (50)$$

Next, we try to find the upper bound of $\mathbb{E}\left[\|A_l\|_F^2\right]$, $\mathbb{E}\left[\|B_l\|_F^2\right]$, and $\mathbb{E}\left[\|C_l\|_F^2\right]$, respectively.

$$\mathbb{E}\left[\|A_l\|_F^2\right] = \eta^2\mathbb{E}\left\|\frac{1}{N}\sum_{i=1}^{N}\nabla F_{i,l}(\mathbf{w}_i^{t-1})V_{i,l}^{t-1}(\bar{V}_l^{t-1})^\top\right\|_F^2$$

$$\leq \eta^2\frac{1}{N}\sum_{i=1}^{N}\mathbb{E}\left\|\nabla F_{i,l}(\mathbf{w}_i^{t-1})V_{i,l}^{t-1}(\bar{V}_l^{t-1})^\top\right\|_F^2 \quad (51)$$

$$\leq \eta^2\frac{1}{N}\sum_{i=1}^{N}\mathbb{E}\left[\left\|\nabla F_{i,l}(\mathbf{w}_i^{t-1})\right\|_F^2\cdot\left\|V_{i,l}^{t-1}\right\|_F^2\cdot\left\|(\bar{V}_l^{t-1})^\top\right\|_F^2\right] \leq \eta^2 G^2 \Gamma_v^4$$

Note that $A_l$ and $B_l$ are symmetric, and similarly, we can deduce $\mathbb{E}\left[\|B_l\|_F^2\right] \leq \eta^2 G^2 \Gamma_u^4$. Then, for $C_l$, we have

$$
\begin{aligned}
\mathbb{E}\left[\|C_l\|_F^2\right] &= \eta^4 \mathbb{E}\left\|\left[\frac{1}{N}\sum_{i=1}^N \nabla F_{i,l}(\mathbf{w}_i^{t-1})V_{i,l}^{t-1}\right] \cdot \left[\frac{1}{N}\sum_{i=1}^N (U_{i,l}^{t-1})^\top \nabla F_{i,l}(\mathbf{w}_i^{t-1})\right]\right\|_F^2 \\
&\leq \eta^4 \mathbb{E}\left\|\left[\frac{1}{N}\sum_{i=1}^N \nabla F_{i,l}(\mathbf{w}_i^{t-1})V_{i,l}^{t-1}\right]\right\|_F^2 \cdot \mathbb{E}\left\|\left[\frac{1}{N}\sum_{i=1}^N (U_{i,l}^{t-1})^\top \nabla F_{i,l}(\mathbf{w}_i^{t-1})\right]\right\|_F^2 \\
&\leq \eta^4 \left[\frac{1}{N}\sum_{i=1}^N \mathbb{E}\left\|\nabla F_{i,l}(\mathbf{w}_i^{t-1})V_{i,l}^{t-1}\right\|_F^2\right] \cdot \left[\frac{1}{N}\sum_{i=1}^N \mathbb{E}\left\|(U_{i,l}^{t-1})^\top \nabla F_{i,l}(\mathbf{w}_i^{t-1})\right\|_F^2\right] \leq \eta^4 G^4 \Gamma_u^2 \Gamma_v^2
\end{aligned}
\tag{52}
$$

Combining the result of $A_l$, $B_l$ and $C_l$, we have

$$
\begin{aligned}
\mathbb{E}\left[\sum_{l=\rho+1}^L \left\|\bar{U}_l^t(\bar{V}_l^t)^\top - \bar{U}_l^{t-1}(\bar{V}_l^{t-1})^\top\right\|_F^2\right] &\leq 3\sum_{l=\rho+1}^L \mathbb{E}\left[\|A_l\|_F^2 + \|B_l\|_F^2 + \|C_l\|_F^2\right] \\
&\leq 3\sum_{l=\rho+1}^L \left[\eta^2 G^2 \Gamma_u^4 + \eta^2 G^2 \Gamma_v^4 + \eta^4 G^4 \Gamma_u^2 \Gamma_v^2\right] = 3(L-\rho)\eta^2 G^2(\Gamma_u^4 + \Gamma_v^4 + \eta^2 G^2 \Gamma_u^2 \Gamma_v^2).
\end{aligned}
\tag{53}
$$

Finally, adding Eq.(49) and Eq.(53) to Eq.(48) yields the result of Lemma 5.

**Lemma 6.** *Under Assumptions 1, 2, 3 and 4, at each iteration $t$, it follows that*

$$
\begin{aligned}
&\mathbb{E}\left[\langle\nabla f(\bar{\mathbf{w}}^{t-1}), \bar{\mathbf{w}}^t - \bar{\mathbf{w}}^{t-1}\rangle\right] \\
&\leq \sum_{l=1}^\rho \frac{-\eta}{2}\mathbb{E}\left[\left\|\nabla f_l(\bar{\mathbf{w}}^{t-1})\right\|_F^2\right] + \sum_{l=1}^\rho \frac{-\eta}{2}\mathbb{E}\left[\left\|\frac{1}{N}\sum_{i=1}^N \nabla f_{i,l}(\mathbf{w}_i^{t-1})\right\|_F^2\right] + \frac{\eta}{2}L_s^2 \Lambda \\
&\quad + \sum_{l=\rho+1}^L \frac{\eta^2 - \eta\psi_{uv}^2}{2}\mathbb{E}\left[\left\|\nabla f_l(\bar{\mathbf{w}}^{t-1})\right\|_F^2\right] + 2(L-\rho)L_s^2\eta^3 E^2 G^2(\Gamma_u^2 + \Gamma_v^2) + \frac{1}{2}(L-\rho)\eta^2 G^4 \Gamma_u^2 \Gamma_v^2
\end{aligned}
\tag{54}
$$

**Proof of Lemma 6.** Similarly, we divide the model parameters into full and low-rank parameters as follows

$$
\begin{aligned}
\mathbb{E}\left[\langle\nabla f(\bar{\mathbf{w}}^{t-1}), \bar{\mathbf{w}}^t - \bar{\mathbf{w}}^{t-1}\rangle\right] &= \mathbb{E}\left[\sum_{l=1}^\rho \mathbb{V}(\nabla f_l(\bar{\mathbf{w}}^{t-1}))^\top \cdot \mathbb{V}(\bar{W}_l^t - \bar{W}_l^{t-1})\right] \\
&\quad + \mathbb{E}\left[\sum_{l=\rho+1}^L \mathbb{V}(\nabla f_l(\bar{\mathbf{w}}^{t-1}))^\top \cdot \mathbb{V}(\bar{U}_l^t(\bar{V}_l^t)^\top - \bar{U}_l^{t-1}(\bar{V}_l^{t-1})^\top)\right],
\end{aligned}
\tag{55}
$$

where $\mathbb{V}(\cdot)$ denotes the operator of converting a matrix to a column vector. Next, for the first term, we have

$$
\begin{aligned}
& \mathbb{E}\left[\sum_{l=1}^{\rho} \mathbb{V}(\nabla f_l(\bar{\mathbf{w}}^{t-1}))^\top \cdot \mathbb{V}(\bar{W}_l^t - \bar{W}_l^{t-1})\right] \\
&= \sum_{l=1}^{\rho} (-\eta)\mathbb{E}\left[\mathbb{V}(\nabla f_l(\bar{\mathbf{w}}^{t-1}))^\top \cdot \mathbb{V}(\frac{1}{N}\sum_{i=1}^{N}\nabla F_{i,l}(\mathbf{w}_i^{t-1}))\right] \\
&\overset{(a)}{=} \sum_{l=1}^{\rho} (-\eta)\mathbb{E}\left[\mathbb{V}(\nabla f_l(\bar{\mathbf{w}}^{t-1}))^\top \cdot \mathbb{V}(\frac{1}{N}\sum_{i=1}^{N}\nabla f_{i,l}(\mathbf{w}_i^{t-1}))\right] \\
&\overset{(b)}{=} \sum_{l=1}^{\rho} \frac{-\eta}{2}\mathbb{E}\left[\left\|\mathbb{V}(\nabla f_l(\bar{\mathbf{w}}^{t-1}))\right\|_2^2 + \left\|\mathbb{V}(\frac{1}{N}\sum_{i=1}^{N}\nabla f_{i,l}(\mathbf{w}_i^{t-1}))\right\|_2^2\right] \\
&\quad + \sum_{l=1}^{\rho} \frac{\eta}{2}\mathbb{E}\left[\left\|\mathbb{V}(\nabla f_l(\bar{\mathbf{w}}^{t-1})) - \mathbb{V}(\frac{1}{N}\sum_{i=1}^{N}\nabla f_{i,l}(\mathbf{w}_i^{t-1}))\right\|_2^2\right] \\
&\overset{(c)}{=} \sum_{l=1}^{\rho} \frac{-\eta}{2}\mathbb{E}\left[\left\|\nabla f_l(\bar{\mathbf{w}}^{t-1})\right\|_F^2 + \left\|\frac{1}{N}\sum_{i=1}^{N}\nabla f_{i,l}(\mathbf{w}_i^{t-1})\right\|_F^2\right] \\
&\quad + \sum_{l=1}^{\rho} \frac{\eta}{2}\mathbb{E}\left[\left\|\nabla f_l(\bar{\mathbf{w}}^{t-1}) - \frac{1}{N}\sum_{i=1}^{N}\nabla f_{i,l}(\mathbf{w}_i^{t-1})\right\|_F^2\right] \\
&\overset{(d)}{\leq} \sum_{l=1}^{\rho} \frac{-\eta}{2}\mathbb{E}\left[\left\|\nabla f_l(\bar{\mathbf{w}}^{t-1})\right\|_F^2\right] + \sum_{l=1}^{\rho} \frac{-\eta}{2}\mathbb{E}\left[\left\|\frac{1}{N}\sum_{i=1}^{N}\nabla f_{i,l}(\mathbf{w}_i^{t-1})\right\|_F^2\right] + \frac{\eta}{2}L_s^2\Lambda,
\end{aligned}
\tag{56}
$$

where (a) follows from $f_i(\mathbf{w}) \triangleq \mathbb{E}_{\xi_i \in \mathbb{D}_i}[F_i(\mathbf{w}, \xi_i)]$. (b) follows from $-\mathbf{a}^\top \mathbf{b} = \frac{1}{2}\left[\|\mathbf{a} - \mathbf{b}\|_2^2 - \|\mathbf{a}\|_2^2 - \|\mathbf{b}\|_2^2\right]$ for two column vectors $\mathbf{a}$ and $\mathbf{b}$. (c) follows from $\|\mathbb{V}(X)\|_2^2 = \|X\|_F^2$ for any matrix $X$. (d) follows from Eq.(57).

$$
\begin{aligned}
& \sum_{l=1}^{\rho} \mathbb{E}\left[\left\|\nabla f_l(\bar{\mathbf{w}}^{t-1}) - \frac{1}{N}\sum_{i=1}^{N}\nabla f_{i,l}(\mathbf{w}_i^{t-1})\right\|_F^2\right] \\
&= \sum_{l=1}^{\rho} \mathbb{E}\left[\left\|\frac{1}{N}\sum_{i=1}^{N}\nabla f_{i,l}(\bar{\mathbf{w}}^{t-1}) - \frac{1}{N}\sum_{i=1}^{N}\nabla f_{i,l}(\mathbf{w}_i^{t-1})\right\|_F^2\right] \\
&\overset{(a)}{\leq} \frac{1}{N}\sum_{i=1}^{N}\mathbb{E}\left[\sum_{l=1}^{\rho}\left\|\nabla f_{i,l}(\bar{\mathbf{w}}^{t-1}) - \nabla f_{i,l}(\mathbf{w}_i^{t-1})\right\|_F^2\right] \\
&\overset{(b)}{\leq} \frac{1}{N}\sum_{i=1}^{N}\mathbb{E}\left[\sum_{l=1}^{L}\left\|\nabla f_{i,l}(\bar{\mathbf{w}}^{t-1}) - \nabla f_{i,l}(\mathbf{w}_i^{t-1})\right\|_F^2\right] \\
&\overset{(c)}{\leq} \frac{1}{N}\sum_{i=1}^{N} L_s^2\mathbb{E}\left[\left\|\bar{\mathbf{w}}^{t-1} - \mathbf{w}_i^{t-1}\right\|_F^2\right] \\
&\overset{(d)}{\leq} L_s^2\Lambda,
\end{aligned}
\tag{57}
$$

where (a) follows from the Minkowski inequality. (b) follows from $\rho \leq L$. (c) follows from the smoothness of $f$. (d) follows from the result of Lemma 4.

For the second term, we have

$$
\mathbb{E}\left[\sum_{l=\rho+1}^{L} \mathbb{V}(\nabla f_l(\bar{\mathbf{w}}^{t-1}))^\top \cdot \mathbb{V}(\bar{U}_l^t(\bar{V}_l^t)^\top - \bar{U}_l^{t-1}(\bar{V}_l^{t-1})^\top)\right]
$$

$$
= \sum_{l=\rho+1}^{L} (-\eta)\mathbb{E}\left[\mathbb{V}(\nabla f_l(\bar{\mathbf{w}}^{t-1}))^\top \cdot \mathbb{V}(\frac{1}{N}\sum_{i=1}^{N}\nabla F_{i,l}(\mathbf{w}_i^{t-1})V_{i,l}^{t-1}(\bar{V}_l^{t-1})^\top)\right]
$$

$$
+ \sum_{l=\rho+1}^{L} (-\eta)\mathbb{E}\left[\mathbb{V}(\nabla f_l(\bar{\mathbf{w}}^{t-1}))^\top \cdot \mathbb{V}(\frac{1}{N}\sum_{i=1}^{N}\bar{U}_l^{t-1}(U_{i,l}^{t-1})^\top\nabla F_{i,l}(\mathbf{w}_i^{t-1}))\right]
$$

$$
+ \sum_{l=\rho+1}^{L} \eta^2\mathbb{E}\left[\mathbb{V}(\nabla f_l(\bar{\mathbf{w}}^{t-1}))^\top \cdot \mathbb{V}(\left[\frac{1}{N}\sum_{i=1}^{N}\nabla F_{i,l}(\mathbf{w}_i^{t-1})V_{i,l}^{t-1}\right] \cdot \left[\frac{1}{N}\sum_{i=1}^{N}(U_{i,l}^{t-1})^\top\nabla F_{i,l}(\mathbf{w}_i^{t-1})\right])\right]
$$

$$
\overset{(a)}{=} \sum_{l=\rho+1}^{L} (-\eta)\,\underbrace{\mathbb{E}\left[\mathbb{V}(\nabla f_l(\bar{\mathbf{w}}^{t-1})\bar{V}_l^{t-1})^\top \cdot \mathbb{V}(\frac{1}{N}\sum_{i=1}^{N}\nabla f_{i,l}(\mathbf{w}_i^{t-1})V_{i,l}^{t-1})\right]}_{A_l}
$$
(58)

$$
+ \sum_{l=\rho+1}^{L} (-\eta)\,\underbrace{\mathbb{E}\left[\mathbb{V}((\bar{U}_l^{t-1})^\top\nabla f_l(\bar{\mathbf{w}}^{t-1}))^\top \cdot \mathbb{V}(\frac{1}{N}\sum_{i=1}^{N}(U_{i,l}^{t-1})^\top\nabla f_{i,l}(\mathbf{w}_i^{t-1}))\right]}_{B_l}
$$

$$
+ \sum_{l=\rho+1}^{L} \eta^2\,\underbrace{\mathbb{E}\left[\mathbb{V}(\nabla f_l(\bar{\mathbf{w}}^{t-1}))^\top \cdot \mathbb{V}(\left[\frac{1}{N}\sum_{i=1}^{N}\nabla F_{i,l}(\mathbf{w}_i^{t-1})V_{i,l}^{t-1}\right] \cdot \left[\frac{1}{N}\sum_{i=1}^{N}(U_{i,l}^{t-1})^\top\nabla F_{i,l}(\mathbf{w}_i^{t-1})\right])\right]}_{C_l},
$$

where (a) follows from $f_i(\mathbf{w}) \triangleq \mathbb{E}_{\xi_i \in \mathbb{D}_i}[F_i(\mathbf{w}, \xi_i)]$ and $\mathbf{a}^\top(\mathbf{bc}) = (\mathbf{ab}^\top)^\top\mathbf{c}$ for column vectors $\mathbf{a}$, $\mathbf{b}$ and $\mathbf{c}$.

$$
-A_l \leq \underbrace{-\frac{1}{2}\mathbb{E}\left[\left\|\nabla f_l(\bar{\mathbf{w}}^{t-1})\bar{V}_l^{t-1})^\top\right\|_F^2\right]}_{A_{l,1}} + \underbrace{\frac{1}{2}\mathbb{E}\left[\left\|\nabla f_l(\bar{\mathbf{w}}^{t-1})\bar{V}_l^{t-1})^\top - \frac{1}{N}\sum_{i=1}^{N}\nabla f_{i,l}(\mathbf{w}^{t-1})V_{i,l}^{t-1})^\top\right\|_F^2\right]}_{A_{l,2}}
$$
(59)

$$
-B_l \leq \underbrace{-\frac{1}{2}\mathbb{E}\left[\left\|(\bar{U}_l^{t-1})^\top\nabla f_l(\bar{\mathbf{w}}^{t-1})\right\|_F^2\right]}_{B_{l,1}} + \underbrace{\frac{1}{2}\mathbb{E}\left[\left\|(\bar{U}_l^{t-1})^\top\nabla f_l(\bar{\mathbf{w}}^{t-1}) - \frac{1}{N}\sum_{i=1}^{N}(U_{i,l}^{t-1})^\top\nabla f_{i,l}(\mathbf{w}_i^{t-1})\right\|_F^2\right]}_{B_{l,2}}
$$
(60)

Further, we have

$$
A_{l,1} + B_{l,1} = -\frac{1}{2}\mathbb{E}\left[\left\|\nabla h_l(\bar{\mathbf{x}}^{t-1})\right\|_F^2\right] \overset{(a)}{\leq} -\frac{\psi_{uv}^2}{2}\left\|\nabla f_l(\bar{\mathbf{w}}^{t-1})\right\|_F^2,
$$

$$
A_{l,2} + B_{l,2} = \frac{1}{2}\mathbb{E}\left[\left\|\nabla h_l(\bar{\mathbf{x}}^{t-1}) - \frac{1}{N}\sum_{i=1}^{N}\nabla h_{i,l}(\mathbf{x}_i^{t-1})\right\|_F^2\right] = \frac{1}{2}\mathbb{E}\left[\left\|\frac{1}{N}\sum_{i=1}^{N}\left[\nabla h_{i,l}(\bar{\mathbf{x}}^{t-1}) - \nabla h_{i,l}(\mathbf{x}_i^{t-1})\right]\right\|_F^2\right]
$$
(61)

$$
\overset{(b)}{\leq} \frac{1}{2N}\sum_{i=1}^{N}\mathbb{E}\left[\left\|\nabla h_{i,l}(\bar{\mathbf{x}}^{t-1}) - \nabla h_{i,l}(\mathbf{x}_i^{t-1})\right\|_F^2\right] \overset{(c)}{\leq} \frac{L_s^2}{2N}\sum_{i=1}^{N}\mathbb{E}\left[\left\|\bar{\mathbf{x}}_l^{t-1} - \mathbf{x}_{i,l}^{t-1}\right\|_F^2\right]
$$

$$
\overset{(d)}{\leq} 2L_s^2\eta^2 E^2 G^2(\Gamma_u^2 + \Gamma_v^2),
$$

where (a) follows from Lemma 1. (b) follows from the Minkowski inequality. (c) follows from the smoothness of $h$. (d)

follows from Lemma 3. Next, for $C_l$, we have

$$
\begin{aligned}
C_l &\leq \frac{1}{2}\mathbb{E}\left[\left\|\nabla f_l(\bar{\mathbf{w}}^{t-1})\right\|_F^2\right] + \frac{1}{2}\mathbb{E}\left[\left\|\left[\frac{1}{N}\sum_{i=1}^N \nabla F_{i,l}(\mathbf{w}_i^{t-1})V_{i,l}^{t-1}\right]\cdot\left[\frac{1}{N}\sum_{i=1}^N (U_{i,l}^{t-1})^\top \nabla F_{i,l}(\mathbf{w}_i^{t-1})\right]\right\|_F^2\right] \\
&\leq \frac{1}{2}\mathbb{E}\left[\left\|\nabla f_l(\bar{\mathbf{w}}^{t-1})\right\|_F^2\right] + \frac{1}{2}G^4\Gamma_u^2\Gamma_v^2
\end{aligned}
\tag{62}
$$

Combining the results of $A_l$, $B_l$ and $C_l$, we have

$$
\begin{aligned}
&\mathbb{E}\left[\sum_{l=\rho+1}^L \mathbb{V}(\nabla f_l(\bar{\mathbf{w}}^{t-1}))^\top \cdot \mathbb{V}(\bar{U}_l^t(\bar{V}_l^t)^\top - \bar{U}_l^{t-1}(\bar{V}_l^{t-1})^\top)\right] \\
&= \sum_{l=\rho+1}^L \eta(-A_l - B_l) + \sum_{l=\rho+1}^L \eta^2 C_l \\
&\leq \sum_{l=\rho+1}^L \eta(A_{l,1} + B_{l,1} + A_{l,2} + B_{l,2}) + \sum_{l=\rho+1}^L \eta^2 C_l \\
&\leq \sum_{l=\rho+1}^L \frac{\eta^2 - \eta\psi_{uv}^2}{2}\mathbb{E}\left[\left\|\nabla f_l(\bar{\mathbf{w}}^{t-1})\right\|_F^2\right] + 2(L-\rho)L_s^2\eta^3 E^2 G^2(\Gamma_u^2 + \Gamma_v^2) + \frac{1}{2}(L-\rho)\eta^2 G^4\Gamma_u^2\Gamma_v^2
\end{aligned}
\tag{63}
$$

Finally, adding Eq.(56) and Eq.(63) to Eq.(55) yields the result of Lemma 6.

### C.4. Proof of Theorem 1

**Theorem 1.** *Under Assumptions 1, 2, 3 and 4, let $1 < c < 2$ be a constant and the learning rate satisfy $0 < \eta \leq \min\{(\frac{\psi_{uv}^2}{2})^{\frac{1}{c-1}}, \frac{1}{L_s}, 1\}$, we have*

$$
\frac{1}{T}\sum_{t=1}^T \mathbb{E}\left[\left\|\nabla f(\bar{\mathbf{w}}^{t-1})\right\|_2^2\right] \leq \frac{2}{\eta^c T}(f(\bar{\mathbf{w}}^0) - f(\mathbf{w}^*)) + \mathcal{O}(\eta^{2-c})\left[1 + (L-\rho)\mathcal{O}(\Gamma_u^2\Gamma_v^2) + (L-\rho)\mathcal{O}(\Gamma_u^4 + \Gamma_v^4)\right]
\tag{64}
$$

*where $\mathbf{w}^*$ is the optimal parameters, $\Gamma_u^2 \triangleq \min\{2\epsilon_u^2 + 2\eta^2 S^2 G^2\kappa_v^2, \kappa_u^2\}$ and $\Gamma_v^2 \triangleq \min\{2\epsilon_v^2 + 2\eta^2 S^2 G^2\kappa_u^2, \kappa_v^2\}$.*

**Proof of Theorem 1** According to Assumption 1, we have

$$
\mathbb{E}\left[\nabla f(\bar{\mathbf{w}}^t)\right] \leq \mathbb{E}\left[\nabla f(\bar{\mathbf{w}}^{t-1})\right] + \mathbb{E}\left[\langle\nabla f(\bar{\mathbf{w}}^{t-1}), \bar{\mathbf{w}}^t - \bar{\mathbf{w}}^{t-1}\rangle\right] + \frac{L_s}{2}\mathbb{E}\left[\left\|\bar{\mathbf{w}}^t - \mathbf{w}_i^t\right\|_2^2\right]
\tag{65}
$$

Using the results of Lemmas 5 and 6, we have

$$
\begin{aligned}
\mathbb{E}\left[\nabla f(\bar{\mathbf{w}}^t)\right] &\leq \mathbb{E}\left[\nabla f(\bar{\mathbf{w}}^{t-1})\right] \\
&\underbrace{+ \sum_{l=1}^\rho \frac{-\eta}{2}\mathbb{E}\left[\left\|\nabla f_l(\bar{\mathbf{w}}^{t-1})\right\|_F^2\right] + \sum_{l=\rho+1}^L \frac{\eta^2 - \eta\psi_{uv}^2}{2}\mathbb{E}\left[\left\|\nabla f_l(\bar{\mathbf{w}}^{t-1})\right\|_F^2\right]}_{A_t} \\
&\underbrace{+ \sum_{l=1}^\rho \frac{-\eta}{2}\mathbb{E}\left[\left\|\frac{1}{N}\sum_{i=1}^N \nabla f_{i,l}(\mathbf{w}_i^{t-1})\right\|_F^2\right] + \sum_{l=1}^\rho \frac{L_s\eta^2}{2}\mathbb{E}\left[\left\|\frac{1}{N}\sum_{i=1}^N \nabla f_{i,l}(\mathbf{w}_i^{t-1})\right\|_F^2\right]}_{B_t} \\
&\underbrace{+ \frac{\eta}{2}L_s^2\Lambda + 2(L-\rho)L_s^2\eta^3 E^2 G^2(\Gamma_u^2 + \Gamma_v^2) + \frac{1}{2}(L-\rho)\eta^2 G^4\Gamma_u^2\Gamma_v^2 + \frac{L_s\eta^2\sigma^2}{2N} + \frac{3L_s}{2}(L-\rho)\eta^2 G^2(\Gamma_u^4 + \Gamma_v^4 + \eta^2 G^2\Gamma_u^2\Gamma_v^2)}_{C}
\end{aligned}
\tag{66}
$$

For $A_t$, $B_t$, we have

$$
\begin{aligned}
A_t &\overset{(a)}{\leq} \sum_{l=1}^{\rho} \frac{-\eta^c}{2} \mathbb{E}\left[\left\|\nabla f_l(\bar{\mathbf{w}}^{t-1})\right\|_F^2\right] + \sum_{l=\rho+1}^{L} \frac{\eta^c - \eta\psi_{uv}^2}{2} \mathbb{E}\left[\left\|\nabla f_l(\bar{\mathbf{w}}^{t-1})\right\|_F^2\right] \\
&= \sum_{l=1}^{L} \frac{-\eta^c}{2} \mathbb{E}\left[\left\|\nabla f_l(\bar{\mathbf{w}}^{t-1})\right\|_F^2\right] + \sum_{l=\rho+1}^{L} \frac{2\eta^c - \eta\psi_{uv}^2}{2} \mathbb{E}\left[\left\|\nabla f_l(\bar{\mathbf{w}}^{t-1})\right\|_F^2\right] \\
&\overset{(b)}{\leq} \sum_{l=1}^{L} \frac{-\eta^c}{2} \mathbb{E}\left[\left\|\nabla f_l(\bar{\mathbf{w}}^{t-1})\right\|_F^2\right] \leq \frac{-\eta^c}{2} \mathbb{E}\left[\left\|\nabla f(\bar{\mathbf{w}}^{t-1})\right\|_F^2\right]
\end{aligned}
$$

$$
B_t = \sum_{l=1}^{\rho} \frac{L_s\eta^2 - \eta}{2} \mathbb{E}\left[\left\|\frac{1}{N}\sum_{i=1}^{N}\nabla f_{i,l}(\mathbf{w}_i^{t-1})\right\|_F^2\right] \overset{(c)}{\leq} 0, \tag{67}
$$

where (a) follows from $0 < \eta \leq 1$ and $1 < c < 2$, thus $-\eta \leq -\eta^c$ and $\eta^2 \leq \eta^c$. (b) follows from $\eta \leq (\frac{\psi_{uv}^2}{2})^{\frac{1}{c-1}}$, and thus $2\eta^c - \eta\psi_{uv}^2 \leq 0$. (c) follows from $\eta \leq \frac{1}{L_s}$, and thus $L_s\eta^2 - \eta \leq 0$. Therefore, we have

$$
\mathbb{E}\left[\nabla f(\bar{\mathbf{w}}^t)\right] \leq \mathbb{E}\left[\nabla f(\bar{\mathbf{w}}^{t-1})\right] + \frac{-\eta^c}{2}\mathbb{E}\left[\left\|\nabla f(\bar{\mathbf{w}}^{t-1})\right\|_F^2\right] + C \tag{68}
$$

Rearranging the above inequality, summing over $t \in \{1, 2, \ldots, T\}$ and then dividing both sides by $\frac{\eta^c}{2T}$ yields

$$
\frac{1}{T}\sum_{t=1}^{T} \mathbb{E}\left[\left\|\nabla f(\bar{\mathbf{w}}^{t-1})\right\|_F^2\right] \leq \frac{2}{\eta^c T}(f(\bar{\mathbf{w}}^0) - f(\mathbf{w}^T)) + \frac{2C}{\eta^c} \leq \frac{2}{\eta^c T}(f(\bar{\mathbf{w}}^0) - f(\mathbf{w}^*)) + \frac{2C}{\eta^c} \tag{69}
$$

Next, rearranging the terms in $C$ yields

$$
\begin{aligned}
C =& 2L_s^2\rho E^2 G^2\eta^3 + \frac{L_s\sigma^2}{2N}\eta^2 \\
&+ (L-\rho)2L_s^2 E^2\eta^3 G^2(\Gamma_u^2 + \Gamma_v^2) \\
&+ (L-\rho)(6L_s^2 E^4\eta^3 + \frac{3}{2}L_s\eta^2 + \frac{1}{2})\eta^2 G^4\Gamma_u^2\Gamma_v^2 \\
&+ (L-\rho)(6L_s^2 E^2\eta + \frac{3}{2}L_s)\eta^2 G^2(\Gamma_u^4 + \Gamma_v^4)
\end{aligned} \tag{70}
$$

Multiplying $C$ by $\frac{2}{\eta^c}$ and analyzing its complexity with respect to $\eta$, we have

$$
\begin{aligned}
\frac{2C}{\eta^c} =& 4L_s^2\rho E^2 G^2\eta^{3-c} + \frac{L_s\sigma^2}{N}\eta^{2-c} + (L-\rho)4L_s^2 E^2\eta^{3-c}G^2(\Gamma_u^2 + \Gamma_v^2) \\
&+ (L-\rho)(12L_s^2 E^4\eta^3 + 3L_s\eta^2 + 1)\eta^{2-c}G^4\Gamma_u^2\Gamma_v^2 + (L-\rho)(12L_s^2 E^2\eta + 3L_s)\eta^{2-c}G^2(\Gamma_u^4 + \Gamma_v^4) \\
\triangleq& \mathcal{O}(\eta^{2-c})\left[1 + (L-\rho)\mathcal{O}(\Gamma_u^4 + \Gamma_v^4)\right]
\end{aligned} \tag{71}
$$

Finally, adding Eq.(71) to Eq.(69) yields the result in Theorem 1.

