# OpenReview forum: "The Panaceas for Improving Low-Rank Decomposition in Communication-Efficient Federated Learning"
_ICML.cc/2025/Conference — ICML 2025 poster_

### Official Review · Reviewer_u9mP · 2025-03-07

**Overall Recommendation:** 4

**Summary:**

This paper reduces communication overhead in FL by enhancing low-rank decomposition techniques. The authors focus on three key issues: what to decompose, how to decompose, and how to aggregate. They propose three techniques respectively: Model Update Decomposition (MUD), Block-wise Kronecker Decomposition (BKD), and Aggregation-Aware Decomposition (AAD). The paper provides both theoretical analysis and empirical studies on benchmark datasets.

## update after rebuttal
My concerns are mainly addressed during rebuttal. Thus, I will keep my positive rating.

**Claims And Evidence:**

The claims made in this paper are well-supported by both theoretical proofs and experimental validation.

**Essential References Not Discussed:**

This paper adequately discusses essential literature related to low-rank decomposition in FL.

**Experimental Designs Or Analyses:**

The authors conduct experiments on four datasets, comparing the proposed methods against traditional low-rank decomposition techniques and sota pruning and quantization methods. Ablation studies and hyperparameter analysis are also conducted.

**Methods And Evaluation Criteria:**

This paper focuses on improving communication efficiency in FL by proposing three techniques that comprehensively improve low-rank decomposition. The proposed methods have been validated on multiple benchmark datasets.

**Other Comments Or Suggestions:**

I suggest the authors provide a more detailed discussion on the rank upper bound of BKD.

**Other Strengths And Weaknesses:**

Strengths:
1. Communication overhead in FL is a critical problem. This paper identifies three key challenges in low-rank decomposition for FL and proposes three novel techniques to address them.
2. The paper provides a rigorous theoretical analysis, proving that FedMUD achieves better convergence than the traditional method.

Weaknesses:
1. The authors claim that BKD achieves $rank(W)\leq\min\{m,n\}$. I suggest including a more detailed explanation.

**Questions For Authors:**

In Section 5.5, the authors discuss the significant impact of initialization on low-rank decomposition performance. I am curious whether traditional low-rank decomposition methods in FL also face this issue. If so, how do they typically determine the initialization size?

**Relation To Broader Scientific Literature:**

NA

**Theoretical Claims:**

The claims are well-supported by the proofs in general.

---

> ### Author Rebuttal · Authors · 2025-04-01
>
> **Hi Reviewer u9mP:**
>
> We sincerely appreciate your valuable feedback. Below, we address each of your comments in detail. For additional experimental results, please refer to the anonymous link: ​**https://anonymous.4open.science/r/fedmud_rebuttal-962F**.
>
> ***Q1: "a more detailed discussion on the upper rank bound of BKD"***
>
> **R1:**
> According to your suggestion, we provide a detailed discussion on the upper rank bound of BKD.
>
> 1. Theoretical Analysis: We leverage the properties of the Kronecker product ($rank(A \otimes B) = rank(A) \cdot rank(B)$) and matrix concatenation ($rank(A|B) \leq rank(A) + rank(B)$). Suppose $A, B \in \mathbb{R}^{a \times b}$, then $W = A \otimes B \in \mathbb{R}^{a^2 \times b^2}$. By the nature of the Kronecker product, $rank(A \otimes B) = rank(A) \cdot rank(B) \leq \min\\{a^2, b^2\\}$, which corresponds to the full rank of a matrix with dimensions $(a^2, b^2)$. Furthermore, BKD can be viewed as the concatenation of multiple Kronecker product matrices. Given that each submatrix retains full rank, the concatenated BKD recovery matrix also achieves full rank.
>
> 2. Experimental Validation: We compare the rank of model updates obtained via matrix multiplication and BKD. The results, presented in **Figure 10** in the above link, demonstrate that BKD produces near full-rank model updates, significantly outperforming standard matrix multiplication.
>
> We will incorporate above disscussion into the paper.
>
> ***Q2: "how do other low-rank decomposition methods handle the initialization of low-rank matrices"***
>
> **R2:**
> Other low-rank decomposition methods, such as FedLMT and FedPara, also suffer from initialization variance.
>
> Hyeon-Woo et al., in FedPara, employ the default He initialization [1] but acknowledge that "investigating initializations appropriate for our model might improve potential instability in our method."
>
> Similarly, Liu et al., in FedLMT, state that "the performance of low-rank models can be boosted through a customized initialization called spectral initialization [2]," which "uses SVD to initialize low-rank model parameters."
>
> In our experiments, to ensure a fair comparison, we adjust the initialization size across different methods. As shown in Figure 4 in our paper, BKD mitigates the sensitivity of the model to initialization variance to some extent. We attribute this to the fact that $Var(AB)=r\cdot Var(A)\cdot Var(B)$ (where $A\in\mathbb{R}^{a\times r}, B\in\mathbb{R}^{r\times b}$, assuming both are i.i.d.), while $Var(A\otimes B)=Var(A)\cdot Var(B)$. Clearly, the variance of the reconstructed matrix using matrix multiplication is also influenced by $r$, making it more unstable.
>
> In this paper, we did not delve into the impact of low-rank matrix initialization on model performance. This is an interesting and meaningful direction that we will continue to pay attention to and conduct possible explorations.
>
> [1] FedPara: Low-Rank Hadamard Product for Communication-Efficient Federated Learning. ICLR 2022.
>
> [2] FedLMT: Tackling System Heterogeneity of Federated Learning via Low-Rank Model Training with Theoretical Guarantees. ICML 2024.
>
> Below, we address a general issue regarding novelty raised by other reviewers, and we hope this will provide you with new insights.
>
>
> ***Q3: "novelty of MUD and difference with LoRA"***
>
> **R3:**
> The core difference between MUD and other model update decompositions is that our model update is for several training rounds, not the entire training process.
> Let's take LoRA as an example to explain the difference and relationship between FedMUD and LoRA in the federated fine-tuning scenario. This also applies to other technologies with similar ideas.
>
> 1. Assuming the pre-trained weight is $W_p$, LoRA will learn $W_p+AB$ during the entire FL training process, where $W_p$ is frozen. $A$ and $B$ are trainable parameters, which represent the model update during the entire FL training process.
> 2. In FedMUD, we learn the model update every $S$ rounds. Reset Interval $s$ is a hyperparameter of fedMUD (the default value is 1), which means that every $S$ rounds, we manually add the model update $AB$ to the pre-trained parameter $W_p$ and reinitialize the submatrix. In this way, the parameters of the final model can be expressed as $W=W_p+\sum_i^{T/s}A_SB_S$, where $T$ is the number of training rounds. By training different low-rank increments in different rounds, we can achieve better accuracy with the same amount of communication.
> 3. When we set $S$ of FedMUD to $\infty$ (or any value larger than the number of communication rounds $T$), which means there is one low-rank increment during the entire training process, FedMUD and FL-LoRA are equivalent. In fact, FL-LoRA is equivalent to FedLMT.
>
> Hopefully the above explanation can clarify for you the difference between how MUD and LoRA and other methods handle model updates.

---

### Official Review · Reviewer_Gugc · 2025-03-10

**Overall Recommendation:** 2

**Summary:**

The paper focuses on enhancing communication efficiency in FL by improving low-rank decomposition techniques.  The authors identify three key issues: what to decompose, how to decompose, and how to aggregate.  To address these, they propose three novel techniques: decompose only model updates, block-wise Kronecker decomposition, and ways the matrices are assembled by the server.  These techniques improve convergence speed and accuracy compared to existing methods.

**Claims And Evidence:**

They are clear but some of the claims are questionable. I disagree with all of the claims regarding the model update decomposition. The original LoRA paper proposes to min f(W+AB) which means that only model updates get a low rank decomposition. Other FL low-rank algorithms also assume the same setting. Unless I'm missing something, the first contribution is not a contribution. To the contrary, it's common practice.

**Essential References Not Discussed:**

Nothing to report.

**Experimental Designs Or Analyses:**

I did. See Methods And Evaluation Criteria for comments.

**Methods And Evaluation Criteria:**

They are okay. Very standard small datasets. It would be great to apply it for find tuning of an LLM.
I wonder why a standard CNN wasn't used (ResNet18 for example) instead of a 'out-of-nowhere' architecture.

**Other Comments Or Suggestions:**

Lines 84 to 88, left column: maximizing the rank is crucial: I'm not sure where this is coming from. In practice and as demonstrated in many paper, low-rank is enough.

**Other Strengths And Weaknesses:**

FedMUD is never formally stated. Based on theory it seems not to include Section 3.2. It's unclear if it includes Section 3.3.
If it includes only Section 3.1, based on my LoRA comment, it questions the contribution of the analytical convergence proof.

I think the idea of using the Kronecker product is a great one. Unfortunately the authors don't do enough with it; it is obfuscated with other aspects such as model update decomposition whose novelty is questionable.

**Questions For Authors:**

1. how is model decomposition different from LoRA and many offsprings?
2. Why not fine tuning of an LLM in experiments?
3. Why not ResNet18 (or a different standard model)?
4. The assumption of bounded weights is questionable. In practice gradient clipping is used because the weights are not bounded (unless being forced to be bounded).

**Relation To Broader Scientific Literature:**

See my other remarks regarding LoRA and other FL low-rank algorithms.

**Theoretical Claims:**

I didn't check the proofs. The statement is a typical convergent statement and thus I does I have fairly high confidence in the proof.

The analysis of the algorithm with Kronecker and aggregation would be much more interesting.
In general, convergence analyses are fairly standard. A more interesting question: with the same number of trainable parameters, compare convergence rates of Kronecker vs standard AB decomposition.

---

> ### Author Rebuttal · Authors · 2025-04-01
>
> **Hi Reviewer Gugc:**
>
> We sincerely appreciate your valuable feedback. Below, we address each of your comments in detail. For additional experimental results, please refer to the anonymous link: **https://anonymous.4open.science/r/fedmud_rebuttal-962F**.
>
> ***Q1: "datasets and models"***
>
> **R1:**
> In response to your question, we have added experiments with other model architectures (ResNet18, ViT) and datasets (TinyImageNet). Specifically, we test ResNet18 on CIFAR-10 and TinyImageNet, and ViT (pretrained on ImageNet) on CIFAR-10. We compared only FedAvg, FedLMT/FedHM, and FedMUD+BKD+AAD. The results (**Figure 8** in the above link) further validate the superiority of our method. By the way, due to limited time, we did not test on LLM. But we conducted experiments on a transformer-based model (ViT) and proved that our method should be applicable to LLM. We will include the complete experimental results in the revised paper.
>
> ***Q2: "FedMUD's formal statement"***
>
> **R2:**
> We have formally defines FedMUD in Section 3.1, line 159: *Federated Learning with Model Update Decomposition (FedMUD)*. Notably, FedMUD refers to using only MUD, excluding the BKD and AAD designs. For better presentation, we will add its formal definition in Section 1 of the revised paper.
>
> ***Q3: "difference between model update decomposition (MUD) and LoRA"***
>
> **R3:**
> Please kindly refer to **R3** from reviewer **u9mP** for our response.
>
> ***Q4: "BKD is obfuscated with other aspects such as MUD"***
>
> **R4:**
> Table 1 presents the accuracy results using FedMUD as the base framework, with BKD and AAD as plugins. The role of BKD is indeed obfuscated with that of MUD or AAD. However, Figure 3 provides an ablation study on the reset interval. Notably, when the reset interval equals the total number of rounds, MUD no longer contributes. In this case, FedMUD+BKD reflects the performance of BKD alone. For further details, please refer to Figure 3 in our paper, which intuitively demonstrates the superiority of BKD over the standard decomposition.
>
> ***Q5: "convergence rates of BKD"***
>
> **R5:**
> The convergence of BKD is indeed a noteworthy problem. Theorem 1 focuses on the convergence of FedMUD (excluding BKD and AAD), demonstrating that FedMUD converges faster than FedLMT. However, comparing the convergence of BKD with methods using matrix multiplication is challenging. In convergence analysis, unifying the number of parameters across different decomposition operators is difficult, making a direct comparison of convergence rates infeasible. We will continue to explore solutions to this problem to provide stronger theoretical support for BKD.
>
> ***Q6: "whether low-rank is enough in practice"***
>
> **R6:**
> In few cases, low-rank approximations are sufficient, such as in LoRA fine-tuning of LLM on small datasets. However, in most scenarios, low-rank methods are primarily used to reduce computational costs, model size, and communication overhead rather than because they are inherently sufficient. Conversely, many approaches aim to achieve higher matrix ranks using a limited number of parameters. Such as FedPara [1] uses Hadamard product to increase matrix rank, whereas MeLoRA [2] enhances the rank of LoRA by arranging smaller matrices along the diagonal.
> Besides, Figure 5 in our paper reveals that as the rank, controlled by the compression ratio, increases, model accuracy improves. This observation confirms that higher ranks contribute to better accuracy. Thus, low-rank approximations are generally insufficient, particularly for communication compression.
>
> [1]FedPara: Low-Rank Hadamard Product for Communication-Efficient Federated Learning, ICLR 2022
> [2]MELoRA: Mini-Ensemble Low-Rank Adapters for Parameter-Efficient Fine-Tuning, ACL 2024
>
> ***Q7: "bounded weights assumption"***
>
> **R7:**
> Your concern regarding the bounded weights assumption is valid. However, FedMUD ensures its reasonableness by periodically reinitializing the low-rank matrices:
>
> 1. First, we assume that the gradients are bounded, which is a reasonable and commonly adopted assumption in convergence analysis, as discussed in lines 245–247 of the paper.
> 2. The bounded weights assumption applies to the low-rank matrices $U$ and $V$ during local training. In FedMUD, $U$ and $V$ are updated via gradient descent for a finite number of steps, after which they are manually added into the base model parameters and reinitialized. This process prevents the unbounded growth of $U$ and $V$.
> 3. To further substantiate the feasibility of this assumption, we provide a theoretical justification. Let $U$ represent a matrix with an initial value $U_0$ of dimension $n$, an upper gradient bound $G$, and $\tau$ gradient descent steps before reinitialization. Then, $U_t = U_0 +\eta\sum_{i=1}^t g_i$, where $t\leq \tau$. Thus, $\Vert U_t\Vert^2\leq (t+1) \left[\Vert U_0 \Vert^2+\eta^2 \sum_{i=1}^t\Vert g_i\Vert^2 \right] \leq (\tau+1)\left[\Vert U_0\Vert^2+\eta^2\tau^2 G^2\right]$.

---

> > ### Comment · Reviewer_Gugc · 2025-04-01
> >
> > Thanks for providing the answers. I have no further questions and comments.

---

> > > ### Author Response · Authors · 2025-04-02
> > >
> > > **Hi, Reviewer Gugc:**
> > >
> > > Thank you again for your time in reviewing our paper. We also sincerely appreciate your acknowledgment of our rebuttal responses. As you indicated no remaining questions and comments, we would be truly grateful if you could reconsider your evaluation score. Your generous reconsideration would mean a lot to our research team.
> > >
> > >
> > > **Follow-up reply (AoE Time: April 6, 2:45 AM)：**
> > >
> > > Hi, Reviewer Gugc! We understand that you have a busy schedule, but we sincerely hope you can reconsider your score in light of our rebuttal. If you still have any concerns, we would greatly appreciate your feedback so we can further improve our work. Thank you again for your time and effort!
> > >
> > > **Follow-up reply (AoE Time: April 7, 1:15 AM):**
> > >
> > > As the rebuttal deadline is approaching, we wanted to kindly check if you might have had a chance to reconsider the evaluation score based on our rebuttal response. If there are any remaining concerns we could address, we would be more than happy to provide additional information. Your feedback is invaluable to us, and we deeply respect your perspective.
> > >
> > > Thank you again for your support throughout this process. We understand your schedule is demanding, and we sincerely appreciate your reconsideration.
> > >
> > > **Final Kind Reminder  (AoE Time: April 8, 3:00 PM)**
> > >
> > > Dear Reviewer Gugc,
> > >
> > > As the rebuttal period is about to close, we wanted to express our sincere gratitude once again for your time and thoughtful feedback throughout this process. Your insights have been invaluable in strengthening our work.
> > >
> > > If you might have a moment to revisit your evaluation score based on our responses, we would be truly grateful. We fully respect your expertise, and even a modest adjustment would mean a great deal to us.
> > >
> > > Of course, we understand how demanding your schedule must be, but want to kindly highlight this final opportunity. Either way,  we deeply appreciate your contributions to improving our paper.

---

### Official Review · Reviewer_fwVG · 2025-03-16

**Overall Recommendation:** 1

**Summary:**

The authors introduce a novel communication-efficient federated learning algorithm that integrates Model Update Decomposition (MUD), Block-wise Kronecker Decomposition (BKD), and Aggregation-Aware Decomposition (AAD). This approach is particularly well-suited for training large neural networks, which commonly involve high-dimensional tensors. They provide a theoretical analysis of the local method applied to a restricted function $h_i(x)$, where $x$ represents the trainable low-rank updates to the neural network parameters. The proposed algorithm is further validated through extensive computational experiments.

**Claims And Evidence:**

The authors effectively support their claims with extensive computational experiments. However, I find Theorem 1 somewhat detached from the main flow of the paper. Although the caption of Figure 1, the discussion on discrepancies between equations (6) and (7), and the introduction of AAD collectively refute the direct aggregation of updates, the proof of Theorem 1 appears to analyze the opposite. If my understanding is correct, this discrepancy is particularly evident in equation (18), which forms the basis of the reviewer's concern. Moreover, the theoretical analysis seems to focus on MUD with truncated SVD rather than with BKD.

**Essential References Not Discussed:**

Related work is adequately described.

**Experimental Designs Or Analyses:**

The experiments appear to be valid.

There is a minor concern regarding the validity of linearizing the stack of convolutional layers in the CNN model used in the experiments. Intuitively, flattening all convolutional kernels and then reshaping them back may introduce significant approximation errors. However, I do not consider this to be a major issue.

**Methods And Evaluation Criteria:**

Proposed methods and evalution criteteria seem adequate.

**Other Comments Or Suggestions:**

1. The abbreviation AAD is introduced on line 74 but is only defined later in the text.
2. It is unclear how BKD is applied in equation (8). Specifically, how is the model update $\Delta W$ distributed between the two summations when combining MUD+BKD+AAD?
3. In line 230, the phrase *"L-smooth with a constant $L_s$"* should be revised to *"$L_s$-smooth."*
4. In line 235, the notation $l$ in $\nabla F_{i, l}$ is ambiguous. Does it refer to the layer number?
5. The algorithm description is missing from the main text and appears only in the appendix, in equation (15).

**Other Strengths And Weaknesses:**

-

**Questions For Authors:**

See above.

**Relation To Broader Scientific Literature:**

The paper falls within the broad area of communication-efficient federated learning (CEFL) algorithms. Unlike approaches that employ well-defined compression operators—whether unbiased or biased—which typically first compute an intermediate update based on the iterate and loss before applying the compression operator, this work belongs to a subfield of CEFL where optimization and compression occur simultaneously. A notable example from this subfield is FedLMT, which directly decomposes the neural network layer matrix rather than the update to it.

**Theoretical Claims:**

No, I did not.

---

> ### Author Rebuttal · Authors · 2025-04-01
>
> **Hi Reviewer fwVG:**
>
> We sincerely appreciate your valuable feedback. Below, we address each of your comments in detail. For additional experimental results, please refer to the anonymous link: ​**https://anonymous.4open.science/r/fedmud_rebuttal-962F**.
>
> ***Q1: "The target object of Theorem 1"***
>
> **R1:** Sorry for the confusion caused and hope to clarify this misunderstanding with the following points:
>
> 1. In this paper, we propose three modules to enhance low-rank decomposition performance: MUD, BKD, and AAD. MUD serves as the foundational framework, and FedMUD refers to the training scheme using only MUD. BKD and AAD are plugins that further improve the rank of model updates and avoid aggregation errors. When using BKD and AAD, we will add corresponding labels as shown in Table 1 in our paper (e.g., +BKD or +AAD).
> 2. Theorem 1 analyzes the convergence of FedMUD (without considering BKD and AAD). Its purpose is to highlight the difference between MUD and full-weight decomposition (i.e., FedLMT vs. FedMUD). By introducing the reset interval $S$ (the number of rounds before adding model update to base weights and then reinitializing low-rank matrices), we establish the relationship between FedLMT and FedMUD to compare their convergence rates. Note that FedLMT is equivalent to FedMUD with $S=\infty$.
> 3. We did not conduct a theoretical analysis of BKD and AAD. Instead, we further added experiments (**Figures 10-11** in the above link) demonstrating their effectiveness in increasing the rank of recovered model updates and avoiding aggregation errors. We will revise the paper to clarify these points and consider integrating Section 4 into Section 3.1 to avoid ambiguity.
>
>
> ***Q2: "concern about reshaping convolutional kernels"***
>
> **R2:** Reshaping the convolutional kernel into a 2D matrix for low-rank decomposition is a method employed by FedLMT and [1], representing a common approach to decomposing convolutional layers. This technique aims to  reduce computational complexity from $\mathcal{O}(k^2 c_{in} c_{out})$ to $\mathcal{O}(kr(c_{in} + c_{out}))$. Therefor, for a fair comparison, we follow their processing approach of convolutional kernels.
>
> [1] Initialization and Regularization of Factorized Neural Layers, ICLR 2021.
>
> ***Q3: "definition of abbreviation AAD"***
>
> **R3:**
> Thank you for raising this point. The definition of AAD is provided in Figure 1(c), though it may not be obvious. We will include a clearer, formal definition of AAD at an appropriate location in Section 1.
>
> ***Q4: "how BKD is applied in Eq.(8)"***
>
> **R4:**
> Eqs (8) and (9) describe the process of recovering weights and aggregating submatrices using only AAD. When combined with MUD, the local weights of client $i$ are given by:
>
> $W_i = W_g + [U_i(\widetilde{V})^\top + \widetilde{U}(V_i)^\top]$
>
> where $i$ is the client index, $W_g$ is the latest global model parameter, and $U_i(\widetilde{V})^\top + \widetilde{U}(V_i)^\top$ represents the local model update. $\widetilde{V}$ and $\widetilde{U}$ are random noise terms, determined by a unified random seed, which are untrainable and identical across all clients.
> When incorporating BKD, the matrix multiplication of $UV^\top$ is replaced by the BKD operator ($\otimes_B$), defined as follows:
>
> $W_i = W_g + [U_i\otimes_B\widetilde{V}  + \widetilde{U}\otimes_BV_i]$
>
> After local training, client $i$ only needs to send the trained $U_i$ and $V_i$ to the server for aggregation, regardless of whether standard matrix multiplication or BKD is used.
>
> ***Q5: "algorithm description"***
>
> **R5:**
> Thank you for your suggestion. We will add a detailed description of our algorithm in the main text, as well as an overall algorithm table for a clearer presentation.
>
> ***Q6: "meaning of $l$ in $\nabla F_{i,l}$"***
>
> **R6:**
> $l$ denotes the network layer, and $i$ is the client index. We will include a detailed and prominent explanation in the paper.
>
> ***Q7: "the expression of $L$-smooth"***
>
> **R7:**
> Thank you for the correction. We will revise the description about $L_s$-smooth in the paper accordingly.

---

### Official Review · Reviewer_W27g · 2025-03-19

**Overall Recommendation:** 4

**Summary:**

This paper introduces three techniques to enhance low-rank decomposition for communication-efficient federated learning (CEFL):

**Model Updates Decomposition** (MUD), **Block-wise Kronecker Decomposition** (BKD), and **Aggregate-Aware Decomposition** (AAD). Each method addresses specific challenges—what to decompose, how to decompose, and how to aggregate. Theoretical analysis is provided to prove convergence, and experiments across multiple datasets (FMNIST, SVHN, CIFAR-10, and CIFAR-100) are conducted, demonstrating superior accuracy compared to baselines.

**Claims And Evidence:**

- Claim 1: MUD reduces information loss by decomposing only model updates rather than full parameters.
  - **Evidence**: Theoretical discussion and experimental results show reduced compression errors and better accuracy.
  - **Concern**: MUD is not novel, as similar ideas exist in LoRA[1] techniques, EvoFed[2], SA-LoRA[3], FFA-LoRA[4], and MAPA[5], which also freeze model parameters and only compress updates. The authors fail to highlight these connections.

- Claim 2: BKD enhances the rank of decomposed matrices, improving information preservation during compression.
  - **Evidence**: Theoretical rank bounds and experimental validation are provided.
  - **Concern**: BKD novelty is limited, as improved rank and parameter efficiency are explored in EvoFed through *reshaping* into single vector and *partitioning*. There is little justification for why this particular approach is better than existing alternatives.

- Claim 3: AAD mitigates implicit aggregation errors introduced by low-rank approximations.
  - **Evidence**: Theoretical formulation and ablation studies validate reduced bias in aggregation.
  - **Concern**: AAD is novel, but it doubles communication costs compared to freezing one matrix. There is no comparison of whether this additional cost is justified by improved performance. (Accuracy per Communication analysis)


[1] LoRA:  Hu, Edward J., et al. "Lora: Low-rank adaptation of large language models. arXiv 2021." arXiv preprint arXiv:2106.09685 (2021).

[2] EvoFed: Rahimi, Mohammad Mahdi, et al. "EvoFed: leveraging evolutionary strategies for communication-efficient federated learning." Advances in Neural Information Processing Systems 36 (2023): 62428-62441.

[3] SA-LoRA: Guo, Pengxin, et al. "Selective Aggregation for Low-Rank Adaptation in Federated Learning." arXiv preprint arXiv:2410.01463 (2024).

[4] FFA-LoRA: Sun, Youbang, et al. "Improving loRA in privacy-preserving federated learning." arXiv preprint arXiv:2403.12313 (2024).

[5] MAPA: Rahimi, Mohammad Mahdi, et al. "Communication-Efficient Federated Learning via Model-Agnostic Projection Adaptation." https://openreview.net/forum?id=rhfOzJzsKN

**Essential References Not Discussed:**

[1] LoRA:  Hu, Edward J., et al. "Lora: Low-rank adaptation of large language models. arXiv 2021." arXiv preprint arXiv:2106.09685 (2021).

[2] EvoFed: Rahimi, Mohammad Mahdi, et al. "EvoFed: leveraging evolutionary strategies for communication-efficient federated learning." Advances in Neural Information Processing Systems 36 (2023): 62428-62441.

**Experimental Designs Or Analyses:**

**Datasets:** FMNIST, SVHN, CIFAR-10, CIFAR-100.
**Data Distribution:** Non-IID and IID settings.
**Metrics:** Test accuracy and convergence speed.
**Comparisons:** Against multiple baseline algorithms under controlled hyperparameters.
**Ablation Studies:** Performed to isolate the effects of MUD, BKD, and AAD, as well as the impacts of reset intervals and initialization values.

**Methods And Evaluation Criteria:**

- **Methods**: Introduction of MUD, BKD, and AAD techniques. Theoretical proofs are provided for convergence and decomposition efficiency.
- **Evaluation Criteria**: Accuracy on multiple datasets under non-IID and IID settings. Baselines include FedHM, FedLMT, FedPara, EF21-P, and FedBAT.

**Other Comments Or Suggestions:**

There are no additional comments or suggestions at this time.

**Other Strengths And Weaknesses:**

**Strengths:**
- Introduction of three complementary techniques that tackle distinct challenges in low-rank decomposition.
- Strong theoretical backing with convergence proofs.
- Comprehensive and well-controlled experimental evaluation.
- Detailed ablation studies enhancing the robustness of the claims.

**Weaknesses:**
- **Lack of Novelty in MUD and BKD:** The core concepts behind MUD and BKD closely align with prior work (e.g., LoRA, EvoFed, MAPA). However, the paper does not sufficiently acknowledge these connections or differentiate its contributions.
- **Insufficient Justification for AAD's Overhead:** AAD effectively doubles communication costs, yet the paper lacks a thorough trade-off analysis to justify this overhead.
- **Scalability Not Demonstrated:** Experiments are confined to relatively small models, with no evaluation on larger-scale tasks (e.g., ImageNet) or with more complex FL architectures.
- **Absence of Computational Cost Analysis:** While communication efficiency is discussed, the computational overhead introduced by the proposed methods on both client and server sides remains unaddressed. This is critical for assessing feasibility in large-scale FL deployments.
- **Hyperparameter Sensitivity:** The methods exhibit high sensitivity to initialization values and reset intervals, necessitating extensive fine-tuning. Moreover, the impact of parameters such as *rr* or *kk* (rank or BKD constant) on training dynamics and accuracy-communication trade-offs is not clearly analyzed.

**Questions For Authors:**

**Overhead Analysis:**
- What are the memory and computational overheads of the proposed methods compared to baseline approaches?

**Communication Costs in AAD:**
- Is the additional communication cost of AAD over Freezing justified across all scenarios, or is it contingent on the dataset and FL setting?

**Scalability Considerations:**
- Can the approach scale to larger datasets and model architectures, such as ImageNet with Transformers?
- How does low-rank factorization perform with architectures beyond MLP and CNN?

**Choice of Parameter _s_:**
- The text suggests that FedMUD reduces to FedLMT when _s = 1_, and also implies that _s = 1_ is often the optimal choice.
- What is the rationale for selecting _s > 1_ in certain scenarios?
- Are there specific advantages to higher _s_ values?

**FedMUD vs. FedLMT Performance:**
- If FedMUD reduces to FedLMT with _s = 1_, why does Table 1 indicate that FedMUD outperforms FedLMT?
- Shouldn’t their accuracies converge in this case?

**Relation To Broader Scientific Literature:**

This work fits within the broader theme of communication-efficient FL. Particularly methods that address reducing communication load through the low-rank representation of the model updates, which can extend to LoRA architecture in FL.

**Theoretical Claims:**

1. **Convergence Guarantees**: Provided under certain and solid assumptions (e.g., L-smoothness, bounded gradients).
2. **Improved Rank and Compression Efficiency**: BKD offers higher rank upper bounds and flexible compression ratios.
3. **Error Reduction**: AAD minimizes bias during aggregation, validated by rigorous mathematical derivation.

**Concern**: While the convergence analysis is provided, additional computational costs are not analyzed, which is critical for understanding scalability.

---

> ### Author Rebuttal · Authors · 2025-04-01
>
> **Hi Reviewer W27g:**
>
> We sincerely appreciate your valuable feedback. Below, we address each of your comments in detail. For additional experimental results, please refer to the anonymous link: ​**https://anonymous.4open.science/r/fedmud_rebuttal-962F**.
>
> ***Q1: "novelty of MUD and difference with LoRA"***
>
> **R1:**
> Please kindly refer to **R3** from reviewer **u9mP** for our response.
>
> ***Q2: "novelty of BKD and difference with EvoFed"***
>
> **R2:**
> There may be some misunderstandings, and we offer the following clarifications:
>
> 1. After carefully studying Evofed, we found that it does not involve reshaping or partitioning vectors, nor does it use low-rank decomposition. In essence, Evofed substitutes the transmission of model parameters with the use of distance similarity between the trained model parameters and a noise-perturbed model population.
>
> 2. The primary focus of BKD is not on reshaping or partitioning vectors, but rather on the innovative application of Kronecker products to enhance the rank of model updates. While the block structure in BKD serves as a partitioning mechanism, its main purpose is to enable dynamic compression strength, rather than being the key factor in improving efficiency.
>
> Therefore, we assert that the innovation of BKD is worthy of recognition.
>
> ***Q3: "AAD's doubles communication costs compared to freezing one matrix."***
>
> **R3:**
> AAD does not double communication costs. The following explanations clarify this:
>
> 1. Compared to standard decomposition ($UV^\top$), AAD ($U\tilde{V}^\top + \tilde{U}V^\top$) does not increase the communication volume. This is because $\tilde{V}$ and $\tilde{U}$ are noise terms determined by random seeds and are not updated. Thus, the communication volume remains #$param(U)$ + #$param(V)$.
>
> 2. The comparison in Table 2 is made under identical communication conditions, meaning that with the same communication volume, AAD outperforms the approach of freezing a matrix.
>
> These points will be emphasized in the revised version of the paper.
>
>
> ***Q4: "additional computational costs."***
>
> **R4:**
> As you noted, while our focus is communication compression, our approach incurs additional computational and memory overhead. We reported the local training time of different methods. Results (**Table 5** in the avove link) show that our method does not excessively increase local training time. On the contrary, the disadvantages of additional computing time overhead can be offset by the benefits of efficient communication and will not affect the acceleration of federated training.
>
> ***Q5: "concern about scalability"***
>
> **R5:**
> Please kindly refer to **R3** from reviewer **Gugc** for our response.
>
> ***Q6: "hyperparameter sensitivity:"***
>
> **R6:**
> Our hyperparameters typically do not require manual tuning. The following clarifications are provided:
>
> 1. Our method introduces three additional hyperparameters: compression strength ($r$ or $k$), reset interval ($s$), and initialization size ($a$).
>
> 2. Compression strength: The values of $r$ and $k$ are determined by the desired compression strength and do not require further adjustment.
>
> 3. Reset interval: Our experiments and analysis indicate that larger values of $s$ lead to lower accuracy. As such, $s$ should be set to 1 by default without modification. To better demonstrate the mechanism of MUD, we include experiments showing the accuracy reduction associated with larger values of $s$. Therefore, $s$ does not need to be treated as a tunable hyperparameter, and there is no sensitivity issue.
>
> 4. Initialization size: While the initialization of sub-matrices affects model accuracy, this issue is inherent in all low-rank methods. As shown in Fig. 4 of our paper, BKD effectively reduces the sensitivity of the model to initialization size. For further details, please refer to **R2** from reviewer **u9mP**.
>
>
> ***Q7: "choice of parameter s and question about FedMUD vs. FedLMT Performance:"***
>
> **R7:**
> FedMUD reduces to FedLMT when $s\geq T$, where $T$ is the number of training rounds, and $s$ denotes the number of rounds for resetting model updates (i.e., adding a submatrix to the base parameters and reinitializing the submatrix). In MUD, the final model parameters are expressed as: $W=W_0+ \sum_i^{T/s} UV^\top$.
> As shown in Figure 3 in our paper, model accuracy generally decreases as $s$ increases, which is consistent with Theorem 1. However, compared to $s=1$, slight improvements in accuracy are occasionally observed for $s=2$ or $s=4$, which can be attributed to data heterogeneity. Specifically, a larger interval allows the low-rank matrices to observe more diverse data before being added to the frozen parameters, thereby mitigating the effects of data heterogeneity. However, as the interval increases further, previously learned knowledge may be forgotten or overwritten, leading to a decline in accuracy.

---

> > ### Comment · Reviewer_W27g · 2025-04-01
> >
> > Thank you for providing the answers.
> > I am satisfied with the answers for **Q1**, **Q2**, **Q4**, **Q6**, and **Q7**.
> >
> > I still think the **R3** and **R5** did not answer my concerns.
> >
> > **Q3**:
> > My concern was not the comparisons of $UV^\top$ to $U\bar{V}^\top + \bar{U}V^\top$, but the comparison of **ADD** to the case where one of the matrices is random like $U\bar{V}^\top$ or $\bar{U}V^\top$.
> > I believe in this case, the communication cost is either $param(U)$ or $param(V)$, while ADD has a communication cost of $param(U) + param(V)$.
> >
> > From your answer and the results in Table 2, I understand that ADD has higher convergence accuracy at the end of the training.
> >
> > However, my question is, given ADD additional communication overhead, compared to $U\bar{V}^\top$ or $\bar{U}V^\top$, does it achieve **higher accuracy per communication**?
> >
> > It would be convincing if the reviewer could provide plots for accuracy per communication or a table indicating the minimum number of rounds and minimum communication cost to reach certain levels of accuracy in each case, such as total communication at 70% and total communication at 80%.
> >
> > I think the only case in which ADD can have higher accuracy per communication would be if ADD converged twice as fast as freezing one matrix.
> >
> > **Q5**:
> > I think there was a mistake; the answer **R3** from the reviewer **Gugc** is unrelated to my questions and, in fact, refers to another response.
> >
> > I believe there should be a time and memory complexity analysis that shows the methodology does not demand high computational resources as the number of clients or model parameters increases.

---

> > > ### Author Response · Authors · 2025-04-02
> > >
> > > Thank you for your thorough review of our response and your timely feedback. We are glad that our response has addressed some of your questions. However, we apologize for not clearly explaining Q3 and for mistakenly linking Q5 to another question's answer. Below, we provide a more detailed response to these two questions.
> > >
> > > ***Q3: "AAD's doubles communication costs compared to freezing one matrix."***
> > >
> > > **R3-(2):**
> > > To address this concern more intuitively, we compare the performance of three settings:
> > > 1. ***AAD*** ($U\tilde{V}^\top+\tilde{U}V^\top$): This setting employs FedMUD+AAD with a 32x compression rate. Here, $\tilde{V}$ and $\tilde{U}$ are randomly initialized and frozen. The communication volume is given by $param(𝑈)+param(𝑉)=param(𝑊)/32$, where $W$ denotes the original weight matrix.
> > > 2. ***AAD-Half*** ($\tilde{U}V^\top$): This setting freezes $\tilde{U}$ and trains $V$.
> > > The shapes of $\tilde{U}$ and $V$ remain the same as in *AAD* (setting 1). However, the communication volume is reduced to $param(𝑉)<param(𝑊)/32\approx param(𝑊)/64$.
> > > 1. ***Freeze*** ($\tilde{A}B^\top$): Similar to *AAD-Half*, this setting also freezes $\tilde{A}$. However, the matrices $\tilde{A}$ and $𝐵$ are larger in shape compared to $\tilde{U}$ and $V$, ensuring that $para(B)=para(W)/32$ (i.e., 32x compression as *AAD*).
> > >
> > > Notably, the relationship between the communication volumes of the three settings follows: ***AAD-Half < AAD = Freeze***.
> > >
> > > We compare these three settings based on two criteria: the same number of training rounds and the same communication volume, as illustrated in Figure 13(a-b) of https://anonymous.4open.science/r/fedmud_rebuttal-962F. **Figure 13(a)** shows that under the same number of training rounds, *AAD-Half* achieves the lowest accuracy due to its strongest compression. In contrast, *AAD* not only significantly outperforms *AAD-Half* but also surpasses *Freeze*.
> > > Similarly, under the same communication volume (**Figure 13(b)**), *AAD* remains superior to both *AAD-Half* and *Freeze*. Notably, *AAD-Half* uses more training rounds (i.e., increased computation and updates) under the same communication volume. As a result, its accuracy surpasses that of *Freeze* but still falls short of *AAD*.
> > >
> > > By the way, the comparison presented in Table 2 of our paper specifically examines *AAD* and *Freeze*.
> > >
> > >
> > >
> > > ***Q5: "concern about scalability"***
> > >
> > > **R5-(2):**
> > > Originally, Q5 was an abbreviation of your comment:
> > > *"Scalability Not Demonstrated: Experiments are confined to relatively small models, with no evaluation on larger-scale tasks (e.g., ImageNet) or with more complex FL architectures."*
> > >
> > > The correct reply should be **R1** from reviewer **Gugc**. We apologize for the mislabeling as R3 and provide the complete response below:
> > >
> > > To verify **scalability**, we have conducted additional experiments with various model architectures (ResNet18, ViT) and datasets (CIFAR-10, TinyImageNet). Specifically, we evaluated ResNet18 on CIFAR-10 and TinyImageNet, and ViT (pretrained on ImageNet) on CIFAR-10. We compared the performance of FedAvg, FedLMT/FedHM, and FedMUD+BKD+AAD. The results (Figure 8 in https://anonymous.4open.science/r/fedmud_rebuttal-962F) further validate the superiority of our method on these models and datasets. The complete experimental results will be included in the revised paper.
> > >
> > >
> > > Regarding your question on **time and memory complexity** analysis, we previously addressed this in **R4** (Table 5 in the above link). There, we compared the local training time of different methods and found that our approach does not introduce much more computational overhead than other low-rank methods.
> > >
> > > To further clarify your concern about **the impact of increasing model parameters on time and memory consumption**, we have now included local training time comparisons for models of different scales in Table 6 in the above link. While Table 5 focused solely on time complexity, Table 6 additionally accounts for memory complexity.
> > >
> > > The results in Table 6 are consistent with those in Table 5, both indicating that our method does not introduce excessive additional computations. Notably, compared to ResNet, our method incurs less additional training time on ViT. This is primarily because the Attention mechanism has inherently higher computational complexity than Convolutional Layers, making the relative increase in computation from our method less impactful on ViT’s overall performance. Finally, we note that time and memory complexity are not directly influenced by the number of clients, so we did not consider this factor in our analysis.
> > >
> > > **We hope these clarifications have fully addressed your concerns. If so, we would be most grateful if you could reconsider your evaluation score of our paper. Your generous reconsideration would mean a lot to our research team.**

---

### Official Review · Reviewer_CShD · 2025-03-20

**Overall Recommendation:** 4

**Summary:**

This paper introduces three novel techniques for Communication Efficient Federated Learning (CEFL) based on low-rank matrix decomposition: Model Updates Decomposition, Block-wise Kronecker Decomposition, and Aggregation-Aware Decomposition, each of which are targetting a specific issue. First, to reduce information loss, a Model Updates Decomposition (FedMUD) approach is developed that  involves factorization of model updates rather than entire model parameters. Second, to maximize the rank of the recovered matrix, a Block-wise Kronecker Decomposition (BKD) approach is proposed that partitions the matrix into blocks and uses Kronecker product decomposition and the block structure enables dynamic compression. Finally, to minimize the effect of compression errors during model aggregation stage, the authors propose an Aggregation-Aware Decomposition (AAD) approach that works by decoupling the multiplication of trainable submatrices. These techniques are complementary and can be applied simultaneously to
achieve optimal performance. The authors provide rigorous theoretical analysis of these methods and provide a comprehensive set of experiments to show that the combinations of the above approaches achieve superior accuracy and faster convergence than baseline methods like FedHM (Yao et al., 2021), FedLMT (Liu et al., 2024), FedPara (Hyeon-Woo et al., 2022), EF21-P (Gruntkowska et al., 2023) and FedBAT (Li et al., 2024b).

**Claims And Evidence:**

The authors do a very comprehensive set of experiments that show speed of convergence with communication volume and training rounds of different combinations of FedMUD + BKD + AAD vs baselines as well as quantify the effect of compression ratios, reset intervals and initialization values on the performance of various combinations of FedMUD, BKD and AAD. The main claim in their paper is that their algorithms do a better job of handling information loss, maximizing the rank of the recovered matrices and reducing the effect of compression errors in the model aggregation stage. What the first set of experiments show is that the speed of convergence (Figure 1) is better with FedMUD + BKD + AAD and that the achieved test accuracy is better than the baselines (Table 1). They do not show that their proposed algorithms reduce information loss or maximize the rank of the recovered matrix or reduce compression errors at the aggregation stage *better than the competitors like FedLMT, FedHM and so on*. What the second set of experiments with compression ratios, reset intervals and initialization values show is that combination of FedMUD + BKD + AAD does best in achieving these goals.

**Essential References Not Discussed:**

One reference that has not been mentioned is this. This work also employs low-rank matrix factorization to do federated learning. It achieves a $\mathcal{O}(1/T)$ rate for the full-gradient and $\mathcal{O}(1/\sqrt{T})$ rate for stochastic gradients.

Dadras, Ali, Sebastian U. Stich, and Alp Yurtsever. "Personalized Federated Learning via Low-Rank Matrix Factorization." OPT 2024: Optimization for Machine Learning. 2024.

**Experimental Designs Or Analyses:**

I went through the description of the experimental setup and analysis of the experiments in the paper and examined the plots and tables to judge soundness/validity of the experiment. As the authors state, the initialization of sub-matrices in the decomposition methods can have a significant effect on the model performance and the submatrices were initialized with values drawn from a uniform distribution $U(-a, a)$ with $a \in \{0.01, 0.05, 0.1, 0.5, 1, 5, 10\}$. What would be interesting is to see how varying this initialization value for elements of $U_l$ and $V_l$ affect the performance of FedMUD + BKD + AAD and other baselines. This is key because they are claiming that their methods are more resilient to performance differences due to differences in initialization. Figure 4 shows an ablation study based on this but the curves are not lined up so it is difficult to make a fair comparison between the different methods and the range of values on the x-axis are different from $a \in \{0.01, 0.05, 0.1, 0.5, 1, 5, 10\}$. So it would be helpful to clarify what the numbers on the x-axis of Figure 4 mean.

**Methods And Evaluation Criteria:**

The choice of benchmark datasets is good and make sense for the problem at hand. Their experiments are comprehensive. It is good to show that speed of convergence of different combinations of FedMUD + BKD + AAD is better than the baselines (Figure 1) and that the achieved test accuracy is better than the baselines (Table 1) across 4 benchmark datasets FMNIST, SVHN, CIFAR-10 and CIFAR-100. They also look at different data distributions (IID, non-IID-1 and non-IID-2) and different compressors: the Rand-K and Top-K compressors. These are the most important experiments. The comprehensive experimental evaluation is why I am inclined towards acceptance of the paper despite the shortcomings I present below.

However, what is missing and really needed are experiments that quantify the effect of compression ratios, reset intervals (where applicable) and initialization values on the performance of various combinations of FedMUD, BKD and AAD *as well as baselines*. In general, we need more additional experiments that backup the claims of the paper: the proposed algorithms FedMUD, BKD and AAD together do a better job of handling information loss, maximizing the rank of the recovered matrices and reducing the effect of compression errors in the model aggregation stage *compared to the baseline approaches*. For instance, what is the experimental evidence that the proposed algorithms maximize the rank of the recovered matrices?

**Other Comments Or Suggestions:**

1. For Equation 43, you have a missing ^ before the "T" so the subscript doesn't render.

**Other Strengths And Weaknesses:**

The main strength of the paper is the extensive experimental evaluation and three different proposed approaches to mitigate various issues associated with employing matrix factorization to train ML models in a federated setting.

**Questions For Authors:**

1. In which experiments do you vary $a \in \{0.01, 0.05, 0.1, 0.5, 1, 5, 10\}$?
2. Can you clarify what the numbers on the x-axis of Figure 4 mean. Are they fixed values used to initialize the matrix elements or are they used as hyperparameters for the distribution from which initial values of the elements of $U_l$ and $V_l$ are set?

**Relation To Broader Scientific Literature:**

Low-rank decomposition (Sainath et al. 2013) is a method for parameter compression in federated learning and works by approximating a  matrix by the product of smaller sub-matrices. More specifically, the server sends a low-rank model tot he clients for training and subsequently receives the optimized low-rank model from them. FedHM (Yao et al. 2021) generate a low-rank model by using truncated SVD to the global model but the SVD algorithm introduces approximation errors causing deviations of the client models from the global model over time. To avoid this issue, FedLMT (Liu et al., 2024) directly trains a pre-decomposed global model, eliminating the need for SVD. FedPara (Hyeon-Woo et al., 2022) uses the Hadamard product to increase the rank of the recovered matrices. (Mei et al., 2022) further improves compression by sharing low-rank matrices across multiple layers. The FedMUD approach is an extension of FedLMT in the sense that it also trains a pre-decomposed global model and FedMUD reduces to FedLMT in the case where we have simply low rank matrix factor updates. The BKD approach uses the same concept as FedPara but with block structures to promote dynamic compression.

**Theoretical Claims:**

I tried to check the correctness of the proofs up to Eq. 43 in the appendix. I don't see any issues so far.

---

> ### Author Rebuttal · Authors · 2025-04-01
>
> **Hi Reviewer CShD:**
>
> We sincerely appreciate your valuable feedback. Below, we address each of your comments in detail. For additional experimental results, please refer to the anonymous link: ​**https://anonymous.4open.science/r/fedmud_rebuttal-962F**.
>
> ***Q1: "experimental evidence of the functionality of the proposed modules"***
>
> **R1:**
> Based on your suggestion, we present experimental evidence supporting the functionality of MUD, BKD, and AAD below:
>
> 1. **MUD Reduces Information Loss:** We apply post-training 'model update decomposition' and 'full-weight decomposition' to the locally trained model (of FedAvG) and compare their validation accuracy. Information loss is measured by the loss in validation accuracy. The experimental setup and results are presented in **Figure 9** (see link). As compression strength increases, the accuracy of the full parameter decomposition drops significantly, even falling below the accuracy achieved before local training, which indicates substantial information loss. In contrast, MUD exhibits minimal accuracy loss.
>
> 2. **BKD Maximizes the Rank of the Recovered Matrix:** We compare the ranks of the model updates obtained by FedMUD and FedMUD+BKD on FMNIST, at the same communication cost. The results, shown in **Figure 10** (see link), indicate that the rank of FedMUD is significantly smaller than the full-rank level, and is constrained by the parameter $r$ derived from the compression ratio. In contrast, FedMUD+BKD, utilizing the Kronecker product, maintains nearly full-rank performance.
>
> 3. **AAD Avoids Aggregation Errors:** We report the errors introduced by aggregating low-rank matrices using methods other than AAD (note that AAD introduces no aggregation error). The specific calculation method and settings are shown in **Figure 11** (see link). The results demonstrate that FedLMT, FedHM, FedMUD, and FedMUD+BKD all exhibit varying degrees of aggregation error. Moreover, it is evident that the aggregation error in the MUD and BKD is considerably smaller than that in the full-weight decomposition model.
>
>
> ***Q2: "which experiments do you vary $a\in\\{0.01,0.05,0.1,0.5,1,5,10\\}$"***
>
> **R2:**
> In our experiments, we initialize the submatrix of low-rank methods using a uniform distribution, $U(-a,a)$, where $a$ controls the initialization size. In Section 5.1, we tune $a$ over the values ${0.01, 0.05, 0.1, 0.5, 1, 5, 10}$ for FedLMT, FedPara, and our method. Specifically, FedLMT uses $a=0.05$, FedDMU and FedPara use $a=0.5$, and FedDMU+BKD uses $a=5$. All other experiments follow this configuration, except for the ablation study of $a$ in Section 5.5 (Figure 5).
>
>
> ***Q3: "the curves in Fig. 4 are not lined up and the meaning of the x-axis is unclear"***
>
> **R3:**
> In our experiments, we initialized the submatrices of the low-rank methods using a uniform distribution, $U(-a,a)$. The x-axis in Figure 4 represents the parameter $a$, which controls the initialization amplitude.
>
> To investigate the impact of initialization, we conducted detailed ablation studies around the optimal value of $a$ for each method. For FedMUD, the optimal value is $a=0.5$, and the ablation interval is $\\{0.1, 0.3, 0.5, 1.0, 3.0\\}$, and for FedMUD+BKD, the optimal value is $a=5$, and the ablation interval is $\\{1, 3, 5, 10, 30\\}$.
>
> As suggested, we further aligned the curves in Figure 4 (refer to **Figure 12** in the above link). The experimental results still show that BKD requires larger values of $a$, which can be attributed to the differing variance effects of Kronecker products and matrix multiplications. Specifically, for $A \in \mathbb{R}^{m \times r}$ and $B \in \mathbb{R}^{r \times n}$, assuming both are i.i.d., we have $Var(AB) = r \cdot Var(A) \cdot Var(B)$ and $Var(A \otimes B) = Var(A) \cdot Var(B)$. Since $r$ affects and amplifies $Var(AB)$, BKD requires a larger initialization range to match the magnitudes.
>
> Additionally, for the same reason, BKD is more robust to changes in $a$, as the variance is no longer influenced by $r$ ($r$ is determined by the compression ratio).
>
>
> ***Q4: "code reproducibility issues"***
>
> **R4:**
> Sorry for the reproducibility issues with our code. Our implementation builds upon the open-source work "FedBAT" (ICML 2024), ensuring replicability. We will update the codebase to include the necessary dependencies and conduct thorough reviews to ensure compatibility across different operating systems and devices.
>
> ***Q5: "typo in Eq.(43)"***
>
> **R5:**
> Thank you for your meticulous review. We will address the typos in the upcoming version.
>
> ***Q6: "One related reference not discussed."***
>
> **R6:**
> Although we covered low-rank decomposition in personalized federated learning in the related work section, we acknowledge the oversight of this valuable work. We will include a discussion of it in the revised version.

---

> > ### Comment · Reviewer_CShD · 2025-04-03
> >
> > Thank you for adding the additional experiments, clarifications and updates. I believe that this merits an increase in my score from 3 to 4.

---

> > > ### Author Response · Authors · 2025-04-03
> > >
> > > Thank you so much for taking the time to review our paper and for your helpful suggestions. We truly appreciate you recognizing our responses and raising your score. Your feedback made a big difference in improving our work!

---

### Decision · Program_Chairs · 2025-05-01

**Decision:**

Accept (poster)

**Comment:**

Three low-rank decomposition methods are suggested to reduce communication overhead in FL: model update decomposition, block-wise Kronecker decomposition, and aggregation-aware decomposition. These methods address different issues and can be used together. Model updates decomposition factorizes model updates rather than entire model parameters to minimize information loss. Block-wise decomposition enables high-rank recovery with a relatively small number of parameters, promoting dynamic compression. Aggregation-aware decomposition reduces compression errors during model aggregation by decoupling multiplication using a trainable matrix. Experiments are extensive, and theoretical supports including convergence analysis are convincing. On the down side, even as the core concepts behind MUD and BKD closely align with prior work (e.g., LoRA, EvoFed, MAPA), the paper does not sufficiently acknowledge the connections or differentiate its contributions. Overall, the strengths and merits outweigh the shortcomings.